# AutoData: A Multi-Agent System
# for Open Web Data Collection

**Tianyi Ma[1][†], Yiyue Qian[3][†][*], Zheyuan Zhang[1], Zehong Wang[1], Xiaoye Qian[3][*], Feifan Bai[4],**
**Yifan Ding[3][†][*], Xuwei Luo[5], Shinan Zhang[3][*], Keerthiram Murugesan[6],**
**Chuxu Zhang[2], and Yanfang Ye[1][‡]**
tma2@nd.edu, yyqian5@gmail.com, {zzhang42, zwang43}@nd.edu,
qianxiaoye1993@hotmail.com, xiaobai@uw.edu
dyf0125@gmail.com, luo446@purdue.edu, zhangshinan@gmail.com,
Keerthiram.Murugesan@ibm.com, chuxu.zhang@uconn.edu, yye7@nd.edu
[1]University of Notre Dame, [2]University of Connecticut, [3]Amazon,
[4]University of Washington, [5]Purdue University, [6]IBM Research

## Abstract

The exponential growth of data-driven systems and AI technologies has intensified the demand for high-quality web-sourced datasets. While existing datasets have proven valuable, conventional web data collection approaches face significant limitations in terms of human effort and scalability. Current data-collecting solutions fall into two categories: wrapper-based methods that struggle with adaptability and reproducibility, and large language model (LLM)-based approaches that incur substantial computational and financial costs. To address these challenges, we propose **AutoData**, a novel multi-agent system for **Auto**mated web **Data** collection, that requires minimal human intervention, i.e., only necessitating a natural language instruction specifying the desired dataset. In addition, AutoData is designed with a robust multi-agent architecture, featuring a novel oriented message hypergraph coordinated by a central task manager, to efficiently organize agents across research and development squads. Besides, we introduce a novel hypergraph cache system to advance the multi-agent collaboration process that enables efficient automated data collection and mitigates the token cost issues prevalent in existing LLM-based systems. Moreover, we introduce Instruct2DS, a new benchmark dataset supporting live data collection from web sources across three domains: academic, finance, and sports. Comprehensive evaluations over Instruct2DS and three existing benchmark datasets demonstrate AutoData's superior performance compared to baseline methods. Case studies on challenging tasks such as picture book collection and paper extraction from surveys further validate its applicability. Our source code and dataset are available at here.

## 1 Introduction

Data is the fuel that powers modern data-centric intelligence systems [1, 2]. State-of-the-art AI models rely heavily on high-quality data to learn patterns, forecast trends, and self-optimize. As the World Wide Web hosts an unparalleled breadth of real-world signals, it has become the default reservoir for large-scale data acquisition. Landmark web-sourced data [3–10] have already accelerated research in various domains, such as natural language processing and computer vision.

---

[†]Both authors contributed equally.
[‡]Corresponding author.
[*]The work is not related to the position at the corresponding institution.

39th Conference on Neural Information Processing Systems (NeurIPS 2025).

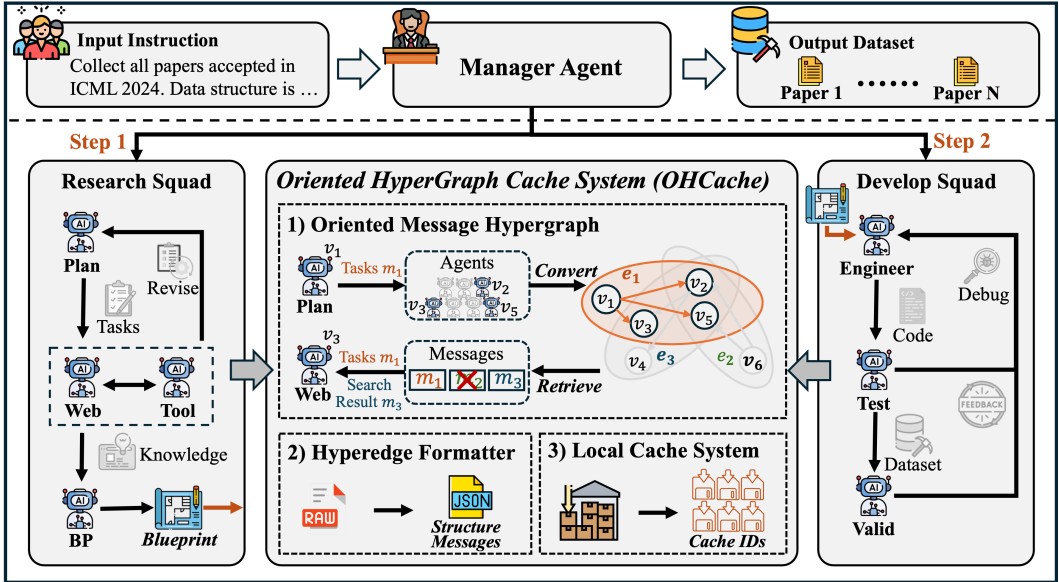

Figure 1: **The overall framework of AutoData.** With the input instruction, agents in the research squad collaborate to generate a development blueprint by browsing the web pages. Afterward, the development squad builds the program based on the blueprint and executes the program to obtain the desired dataset. To ensure efficient and effective multi-agent collaboration, we introduce a novel oriented hypergraph cache system, called OHCache, for information sharing. As shown in the figure, plan agent $v_1$ sends the message $m_1$ to agents $v_2, v_3,$ and $v_5$, resulting in an oriented hyperedge $e_1$. Next, the web agent ($v_3$) retrieves messages $m_1$ and $m_3$ from the oriented message hypergraph $\vec{\mathcal{G}}$ for decision making. In addition, we design a hyperedge formatter to formalize the agent messages and a local cache system to store valuable artifacts for subsequent agents to retrieve in on-demand manners.

Yet demand for richer, domain-specific, and ever-larger datasets is exploding [11]. Conventional web-scraping pipelines buckle under three bottlenecks: (i) Wrapper-based crawlers are brittle, i.e., each new site or layout demands time-consuming, manual rule engineering. (ii) LLM-based scrapers offload the parsing logic to proprietary models but incur substantial temporal and financial overhead. (iii) REST-API harvesters still require human experts to interpret each endpoint parameter and limit by rate rules. The community therefore faces a pivotal question: *"How can we build an end-to-end fully automatic open web data collection system that simultaneously maximizes coverage, accuracy, and efficiency?"*

In response, inspired by the rapid advancement of AI agents that equip LLMs with task-specific tools [12–14], we introduce a novel multi-agent system for **Auto**matic **Data** collection from web sources, called **AutoData**. AutoData orchestrates two specialized agent squads, i.e., research and development, under a central task manager (MGR). Specifically, upon receiving a data collection instruction, specialized agents in the research squad break down the instruction into small steps, follow the steps to extract knowledge from web sources, and design development blueprint. Afterward, the development squad converts the blueprint into executable code, runs the program, and validates the collected data. Unlike existing multi-agent systems for general tasks that ignore the extensive token costs for open web data collection tasks, we introduce a novel **O**riented **H**yperGraph **Cache** System, called **OHCache**, for effective and cost-efficient multi-agent collaboration in open web data collection tasks. OHCache consists of (i) An oriented message hypergraph that models inter-agent message flow as oriented hyperedges. (ii) An oriented hyperedge formatter to enforce the structured communication schema and hyperedge message accumulation. (iii) A local cache system to store reusable artifacts for agents to retrieve on demand.

In addition to the system contribution, we identify an absence of benchmark datasets to evaluate the performance of models for open web data collection tasks. The most relevant benchmark datasets [15, 16], i.e., web information extraction tasks, are merely crawled on static and archived web pages, failing to test on open web data collection. In light of this, we introduce a newly collected

**Instruct**ion to **D**ata**S**et benchmark called Instruct2DS, the first open web data collection benchmark, spanning three domains, i.e., Academic, Finance, and Sport.

Extensive experiments on Instruct2DS and three existing benchmark datasets demonstrate the effectiveness and efficiency of AutoData. Moreover, two additional case studies further highlight its applicability and plug-and-play adaptability. We summarize our contributions as follows:

- **Novel Multi-Agent System**: We develop a fully automatic multi-agent system, consisting of eight specialized agents and a novel oriented hypergraph cache system for efficient and effective multi-agent collaboration, that turns a one-sentence data requirement into a ready-to-use dataset.
- **New Benchmark Dataset**: We introduce a newly collected benchmark Instruct2DS that covers three domains, i.e., Academic, Finance, and Sport. To the best of our knowledge, this is the **first** benchmark dataset to evaluate the performance of models for open web data collection tasks.
- **Comprehensive Evaluation**: We conduct comprehensive experiments on AutoData and baseline methods over Instruct2DS and three existing benchmark datasets, i.e., SWDE, EXTENDED WSDE, and HUMANEVAL. The experiment results demonstrate the effectiveness, efficiency, and applicability of AutoData in open web data collection tasks.

## 2 Preliminary

**Definition 2.1. Oriented Hypergraph.** Let $\mathcal{G} = \{\mathcal{V}, \mathcal{E}\}$ denote a hypergraph, where $\mathcal{V} = \{v_1, \ldots, v_N\}$ is the set of nodes and $\mathcal{E} = \{e_1, \ldots, e_M\}$ is the set of hyperedges. Unlike pairwise edges in graphs [17–19], each hyperedge $e_j \in \mathcal{E}$ connects multiple nodes and represents complex interactions among nodes. In an oriented hypergraph $\vec{\mathcal{G}} = \{\mathcal{V}, \vec{\mathcal{E}}\}$, each hyperedge $\vec{e}_j \in \vec{\mathcal{E}}$ contains two subsets of nodes $\vec{e}_j = (S, T)$, where the subset $S \subseteq \mathcal{V}$ contains the source nodes and the subset $T \subseteq \mathcal{V}$ contains the target nodes to depict a direction among nodes in $\vec{e}_j$.

**Problem 1.** *Open Web Data Collection. Given a data instruction* $\boxed{\text{Instruct}}$ *in natural languages, the objective is to design a fully automated system* $f(\boxed{\text{Instruct}}, \mathcal{W}) = \boxed{\text{DS}}$ *that can effectively and efficiently collect data* $\boxed{\text{DS}}$ *from open web sources* $\mathcal{W}$ *that meet the requirements in* $\boxed{\text{Instruct}}$.

## 3 AutoData: Multi-Agent System for Web Data Collection

AutoData is a multi-agent system (MAS) that is capable of open web data collection through not only web crawling but also REST API calls [20], effectively and efficiently. Specifically, AutoData consists of eight specialized agents that are further divided into the *research squad* and the *development squad* to perform web data collection workflow collaboratively. The design of AutoData is motivated by the traditional web data collection process [21–24] in which the individual first distills data collection logic from web sources and extracts the logic, then codes and executes the program to obtain the datasets. In this section, we first discuss the specialized agents in AutoData. Afterward, we present details of our novel multi-agent collaboration mechanism called the oriented hypergraph cache system (OHCache), which is designed to optimize agent synergy and deliver unparalleled operational efficiency. The overall framework is illustrated in Figure 1.

### 3.1 Specialized Agents

Recent advances underscore the remarkable efficacy of multi-agent collaboration, particularly when agents possess diverse skills and domain expertise, in tackling complex tasks, such as software development [25–28], science debate [29, 30], and recommendation systems [31, 32]. Inspired by this, we design AutoData, which comprises eight specialized agents strategically organized into two synergistic squads, i.e., research and development, coordinated by a central manager agent (MGR), for open web data collection through adaptive agent interaction.

**Research Squad.** The research squad, denoted as $\mathcal{V}_{\text{res}}$, is designed to collaboratively extract, synthesize, and operationalize knowledge critical for web data collection workflows. This group consists of four specialized agents: (i) the plan agent $v_{\text{plan}}$, (ii) the web agent $v_{\text{web}}$, (iii) the tool agent $v_{\text{tool}}$, and (iv) the blueprint agent $v_{\text{bp}}$. Each agent is entrusted with a distinct, complementary role. Specifically, the plan agent $v_{\text{plan}}$ decomposes the overarching data collection objective into granular, actionable steps. The web agent $v_{\text{web}}$ autonomously navigates diverse web sources to extract pivotal

knowledge and procedural logic. The tool agent $v_{\text{tool}}$ leverages advanced utilities (e.g., Google search, file conversion, and HTML cleaner) to augment the squad's operational capabilities. Moreover, the blueprint agent $v_{\text{bp}}$ consolidates the acquired insights into a coherent, actionable development blueprint, laying the foundation for downstream execution.

**Development Squad.** Leveraging the comprehensive blueprint synthesized by the research squad, the development squad $\mathcal{V}_{\text{dev}}$ applies its coding ability to program construction, debugging, execution, and data validation. This squad comprises (i) the engineering agent $v_{\text{engr}}$, (ii) the test agent $v_{\text{test}}$, and (iii) the validation agent $v_{\text{val}}$. The engineering agent $v_{\text{engr}}$ implements the data collection pipeline in strict alignment with the prescribed blueprint. The test agent $v_{\text{test}}$ is responsible for debugging and executing the program, while the validation agent $v_{\text{val}}$ designs and executes sophisticated test cases to ensure data integrity and reliability.

**Manager Agent.** Beyond the specialized research and development agents, we introduce a central manager agent (MGR) $v_{\text{mgr}}$, which orchestrates the end-to-end workflow and mediates inter-group coordination, as illustrated in Figure 1. MGR ensures seamless task allocation, progress tracking, and dynamic adaptation across all agent roles.

Collectively, these specialized agents operate as an integrated, adaptive collective to address the multifaceted challenges of open web data collection. Each agent $v_i$ is characterized by a unique profile that encapsulates its objectives, name, and operational constraints, formalized as a prompt $P_i$. Agents adhere to the ReAct paradigm [33], engaging in deliberate reasoning before action to minimize hallucinations and enhance robustness. Formally, the execution of agent $v_i$ at time $t$ is:

$$m_i^t = \text{LLM}(\mathcal{M}_i^t, P_i, \mathcal{F}_i), \tag{1}$$

where $m_i^t$ is the generated output by agent $v_i$ at time $t$ via the function $\text{LLM}(\cdot)$, $\mathcal{M}_i^t$ represents the message history (see Section 3.2), and $\mathcal{F}_i$ specifies the available function tools for agent $v_i$.

### 3.2 Multi-Agent Collaboration

Effective communication is the linchpin of successful multi-agent collaboration [34–37]. However, current MAS paradigms exhibit critical limitations that hinder their practical deployment in open web data collection scenarios: (i) ***Information overload from broadcast communication.*** Traditional MAS approaches often rely on broadcast-style messaging, where every message is disseminated to all agents and each agent processes the entire message history for decision-making. This strategy not only creates significant information overhead but also increases the risk of hallucination and cognitive overload, especially as message histories grow [38]. (ii) ***Unstructured natural language interfaces.*** Most existing MAS frameworks [39–41] utilize unstructured natural language for inter-agent communication. While flexible, this practice makes it challenging for subsequent agents to extract salient information, impeding efficient problem-solving in complex, multi-stage tasks [26]. (iii) ***Propagation of artifact-embedded messages.*** In open web data collection, agents may inadvertently transmit bulky artifacts (e.g., HTML files, raw web data) within messages, cluttering the communication channel and further compounding inefficiency.

To address these challenges, we propose a novel oriented hypergraph cache system, called OHCache, a communication architecture specifically engineered to optimize information flow, structure message content, and manage artifacts efficiently in MAS environments. Specifically, OHCache includes three components: (i) Oriented Message Hypergraph, (ii) Oriented Hyperedge Formatter, and (iii) Local Cache System. The core insight behind OHCache is that naive one-to-one or broadcast communication topologies introduce unnecessary complexity and inefficiency [35, 42]. Moreover, in practice, merely a few agents require access to the same information source. For instance, both the engineering agent $v_{\text{engr}}$ and validation agent $v_{\text{val}}$ depend on the development blueprint generated by the blueprint agent $v_{\text{bp}}$ for their respective roles in program construction and validation. Oriented hyperedges in a hypergraph elegantly capture this multi-recipient communication, enabling targeted information dissemination that reduces message redundancy and mitigates the risk of overload. Overall, OHCache streamlines communication and enhances overall system scalability and precision. Next, we present each component in OHCache in detail.

**Oriented Message Hypergraph.** To overcome broadcast-induced information overload, we introduce the oriented message hypergraph $\vec{\mathcal{G}}$, which enables selective and efficient message routing:

$$\vec{\mathcal{G}} = (\mathcal{V}, \vec{\mathcal{E}}, \mathcal{M}), \vec{e}_j = (S_j, T_j) \in \vec{\mathcal{E}}, \text{ where } S_j, T_j \subseteq \mathcal{V}, S_j \neq \emptyset, T_j \neq \emptyset, \text{ and } S_j \cap T_j = \emptyset. \tag{2}$$

Here, $\mathcal{V}$ represents the set of agent nodes in Section 3.1, i.e., $\mathcal{V}_{res} \cup \mathcal{V}_{dev} \cup \{v_{mgr}\} = \mathcal{V}$ with size $|\mathcal{V}| = 8$, and message hyperedges $\vec{\mathcal{E}}$ depict the information exchange of messages $\mathcal{M}$ among agents. Particularly, a hyperedge $\vec{e}_j = (S_j, T_j) \in \vec{\mathcal{E}}$, corresponding to the message $m_j^t \in \mathcal{M}$ at time $t$, comprises a source node set $S_j \subseteq \mathcal{V}$ and a target node set $T_j \subseteq \mathcal{V}$. The source node set $S_j$ includes the agent responsible for sending the message, which is consistent with size 1, i.e., $\forall \vec{e}_j \in \vec{\mathcal{E}}, |S_j| = 1$. Conversely, the target node set $T_j$ identifies the group of agents designated to receive the message. This design models real-world scenarios where a message from a single agent is relevant to a specific subset of agents. For any agent $v_i$ scheduled to execute at time $t$, its decision context is constructed from the chronological set of messages $\mathcal{M}_i^t$ directed to it:

$$\mathcal{M}_i^t = \{m_j^{t'} | \vec{e}_j = (S_j, T_j) \in \vec{\mathcal{E}}; v_i \in T_j, t' < t\}. \tag{3}$$

Moreover, the oriented message hypergraph $\vec{\mathcal{G}}$ is dynamically augmented as new messages are recorded. In detail, after an agent execution, the generated message $m_i^t$ via Eq. 1 will be incorporated into the oriented message hypergraph $\vec{\mathcal{G}}$ through the oriented hyperedge formatter (discussed below). Notably, message delivery does not trigger immediate agent execution; instead, the MGR $v_{mgr}$ orchestrates the workflow, leveraging the oriented message hypergraph to ensure efficient, sequential information flow.

**Oriented Hyperedge Formatter.** To address the ambiguity of unstructured natural language, we introduce the oriented hyperedge formatter $g(\cdot)$, which standardizes message content and manages hyperedge insertion. When agent $v_i$ generates a message $m_i^t$, the formatter transforms it into a structured message $\hat{m}_i^t$, and inserts the structured message into the hypergraph:

$$\mathcal{M}' = \mathcal{M} \cup \{\hat{m}_i^t\}, \mathcal{E}' = \mathcal{E} \cup \{\vec{e}_i\}, \text{ where} \tag{4}$$

$$\hat{m}_i^t = g_i(m_i^t) \text{ and } \vec{e}_i = (S_i, T_i). \tag{5}$$

The formatter applies agent-specific rules to ensure that all critical information is explicit and machine-interpretable, facilitating downstream processing. Afterward, the oriented hyperedge formatter updates the oriented message hypergraph as $\vec{\mathcal{G}}' = (\mathcal{V}, \vec{\mathcal{E}}', \mathcal{M}')$. In practice, the structured formats of agents are designed based on their roles in the system, and individuals are requested to provide necessary information for the formatter to generate the structured message. Besides, the relationship among the message $m_i^t$, i.e., hyperedge $\vec{e}_i$, is generated based on the role-specific requirement. For instance, for a hyperedge $\vec{e}_i$ that the source node set contains plan agent $v_{plan}$, i.e., $S_i = \{v_{plan}\}$, the target node set is $T_i = \{v_{web}, v_{tool}, v_{bp}, v_{mgr}\}$.

**Local Cache System.** To prevent unnecessary artifact-embedded messages, we devise a local cache system that serves as a special node $v_{lcs}$ in oriented message hypergraph $\vec{\mathcal{G}}$ to store necessary artifacts. For example, when the web agent $v_{web}$ encounters a valuable artifact, i.e., HTML file, that can assist the engineering agent $v_{engr}$ for programming, node $v_{lcs}$ stores the file locally and broadcasts a concise global message containing a cache identifier. The corresponding hyperedge $\vec{e}_j$ is formulated as:

$$\vec{e}_j = (S_j, T_j), \text{ where } S_j = \{v_{lcs}\} \text{ and } T_j = \mathcal{V}/v_{lcs}. \tag{6}$$

This approach ensures that artifacts are accessible to all agents on demand, without polluting the communication channel. Moreover, agents can subsequently retrieve artifacts by referencing the cache ID, enabling efficient artifact-aware collaboration.

# 4 Instruct2DS: Benchmark Dataset for Open Web Data Collection

As previously discussed, there is currently no publicly available benchmark dataset tailored for data collection tasks from open web sources. To address this gap and enable a comprehensive evaluation of AI agent performance, we introduce a novel *instruction to dataset* benchmark, termed Instruct2DS, designed specifically for web-based data collection across three domains: academic, finance, and sports. In this section, we focus on the academic domain as a representative example. More details about Instruct2DS in the other two domains are provided in Appendix A.

## 4.1 Task Definition

The primary objective of Instruct2DS is to assess the performance of models to complete data collection tasks from real-world web sources based on task instructions. We design symbolic

templates for the generation of a diverse set of instruction variants and adjust complexity levels to better explore the robustness capabilities of models in data collection tasks. Specifically, the dataset Instruct2DS contains the following components:

**Database.** The database $\mathcal{D}$ serves as the authoritative data source for constructing the ground truth dataset $\boxed{\text{GT-DS}}$, which is generated according to the specified task instruction $\boxed{\text{Instruct}}$. The resulting ground truth dataset $\boxed{\text{GT-DS}}$ is used as a reference to evaluate the quality and completeness of the dataset $\boxed{\text{DS}}$ collected by the model. In the academic domain, we comprehensively collect all papers from selected conferences to populate the database. Importantly, during the data collection process, MAS methods are provided only with the task instruction $\boxed{\text{Instruct}}$ and have **no access** to the underlying database. Instead, these methods are required to autonomously gather data directly from open web sources. An illustrative example of a paper entry in the database is shown in Figure 2.

**Instruction Templates.** These are predefined templates $\mathcal{T}$ to build task instructions $\boxed{\text{Instruct}}$ for data collection. Figure 2 lists three instruction templates $\mathbf{T}_{\{1,2,3\}}$ for academic paper collection tasks. The placeholder variables $\mathcal{P}$ in templates $\mathbf{T}_*$, e.g., [Conference] and [Year], etc, represent the specific requirements for the tasks.

**Dataset Construction Algorithm.** We further manually build an algorithm $g(\cdot)$ for ground truth dataset construction from the database $\mathcal{D}$. Formally, given an instruction template $\mathbf{T}_i \in \mathcal{T}$, and corresponding placeholder variables $\mathcal{P}$, the ground truth dataset $\boxed{\text{GT-DS}}$ is obtained as:

$$\boxed{\text{GT-DS}} = g(\mathbf{T}_i, \mathcal{P}, \mathcal{D}). \tag{7}$$

For example, feeding the instruction template $\mathbf{T}_1$ with placeholder variables "conference=NeurIPS" and "year=2017", into the algorithm $g(\cdot)$ will result in a ground truth dataset $\boxed{\text{GT-DS}}$ that contains all accepted paper in NeurIPS 2017, and an example of paper is shown in Figure 2.

---

**A Example of Paper Information**

**TITLE:** Attention is All You Need
**AUTHORS:** Ashish Vaswani, Noam Shazeer, Niki Parmar, Jakob Uszkoreit, Llion Jones, Aidan N Gomez, Łukasz Kaiser, Illia Polosukhin
**ABSTRACT:** The dominant sequence transduction models are based on complex recurrent or convolutional neural networks in an encoder and decoder configuration. The best performing such models also connect the encoder and decoder through an attention mechanism. (skipped for clarity.)
**CONFERENCE:** Advances in Neural Information Processing Systems 30 (NIPS 2017)
**ABBR:** NeurIPS
**TRACK:** Main Conference Track
**PAPER_LINK:** Paper Link
**BIBTEX_LINK:** Bibtex Link
**SUPP_LINK**: None

---

**Instruction Templates for Academic Paper Data Collection**

$\mathbf{T}_1$. Collect all papers accepted in [Conference] [Year].
$\mathbf{T}_2$. Collect all papers accepted in [Conference] [Year] [Track].
$\mathbf{T}_3$. Collect all papers accepted in [Conference] from [Start Year] to [End Year].

---

Figure 2: A sample in Instruct2DS and examples of instruction templates $\boxed{\text{Instruct}}$.

## 4.2 Data Collection

To ensure the long-term availability and stability of Instruct2DS, we selected three domains characterized by persistent and accessible data sources: academic, finance, and sports. For the academic domain, we targeted several top-tier conferences and developed custom Python web crawlers to systematically collect all papers directly from official conference websites. Subsequently, researchers performed thorough cross-validation to verify the completeness, quality, and integrity of the resulting

Table 1: Performance comparison with baseline methods over Instruct2DS. Time indicates the period from the task instruction release to starting data collection, i.e., execute crawler program.

| Model | ACADEMIC | | | STOCK | | | SPORT | | | Time (↓) | Expense (↓) |
|---|---|---|---|---|---|---|---|---|---|---|---|
| | F1 | Prec. | Reca. | F1 | Prec. | Reca. | F1 | Prec. | Reca. | Mins | USD ($) |
| Human | 85.57 | 90.63 | 80.81 | 91.66 | 95.62 | 88.95 | 89.50 | 93.40 | 85.46 | 186.98 | – |
| Cline [45] | 83.50 | 89.65 | 77.12 | 91.37 | 95.20 | 87.17 | 87.07 | 91.38 | 84.53 | 83.29 | – |
| Cursor [46] | 84.37 | 88.53 | 81.02 | 90.23 | 94.24 | 86.31 | 88.70 | 92.98 | 84.38 | 71.60 | – |
| Manus [47] | 69.27 | 72.76 | 66.10 | 95.24 | 97.10 | 93.47 | 87.48 | 93.12 | 81.80 | 15.37 | 2.49 |
| AutoAgent [48] | 65.79 | 67.67 | 63.84 | 73.54 | 94.37 | 52.71 | 75.72 | 88.30 | 63.62 | 10.46 | 1.28 |
| OpenManus [49] | 67.39 | 69.54 | 64.40 | 79.75 | 97.30 | 66.85 | 85.74 | 90.21 | 80.45 | 6.84 | 1.08 |
| AutoAgents [50] | 63.42 | 70.80 | 57.34 | 75.52 | 95.45 | 55.81 | 79.83 | 87.50 | 71.48 | 9.12 | 1.36 |
| **AutoData** | **91.85** | **93.74** | **90.51** | **96.75** | **98.37** | **94.17** | **90.14** | **95.63** | **85.28** | **5.58** | **0.57** |

Table 2: The performance comparison with baseline methods over SWDE and EXTENDED SWDE. Exec-P. and Exce-R. denote the execution precision and recall, respectively.

| Model | SWDE | | | | | | EXTENDED SWDE | | | | | |
|---|---|---|---|---|---|---|---|---|---|---|---|---|
| | F1 | Prec. | Reca. | Correct | Exec-P. | Exce-R. | F1 | Prec. | Reca. | Correct | Exce-P. | Exce-R. |
| COT [51] | 76.95 | 87.75 | 79.90 | 61.88 | 12.50 | 7.19 | 65.08 | 85.15 | 68.35 | 56.10 | 2.44 | **7.32** |
| Reflexion [52] | 82.40 | 93.28 | 82.76 | 67.50 | 13.75 | 4.37 | 75.85 | 87.39 | 77.81 | 64.81 | 4.18 | 5.57 |
| Manus [47] | 89.22 | **93.60** | 88.52 | 73.59 | **14.85** | 4.86 | 75.64 | 81.70 | 79.66 | 63.53 | 4.20 | 6.04 |
| AutoScraper [53] | 88.69 | 92.49 | 89.13 | 71.56 | 14.06 | **5.31** | 76.21 | 82.71 | 80.25 | 64.11 | 3.48 | 6.27 |
| **AutoData** | **89.25** | 92.43 | **90.59** | **74.17** | 14.51 | 5.12 | **77.44** | 80.81 | **81.40** | **65.39** | 3.82 | 5.52 |

database. Afterward, to validate the correctness of the dataset construction algorithm, these researchers further code separately, and cross-validate the algorithm by testing the dataset generated by every valid task instruction. This rigorous process guarantees both the reliability and reproducibility of Instruct2DS. More detail about data collection is provided in Appendix A.

## 4.3 Comparison with Existing Works

Instruct2DS introduces a distinctive set of research challenges, advancing the development of MAS methods for real-world web environments. Unlike existing benchmark datasets [15, 16, 43, 44], Instruct2DS stands out in several key dimensions: (i) **Open Web Setting:** Instruct2DS requires MAS agents to interact with live, dynamic web sources for data collection, in contrast to prior datasets that restrict agents to static, locally archived pages. (ii) **Diverse Data Acquisition Modalities:** The benchmark extends beyond traditional web crawling from HTML sources, incorporating tasks that necessitate data collection through REST API calls, thereby broadening the operational scope and testing the versatility of MAS solutions. (iii) **Symbolic Information Extraction:** Instruct2DS supports symbolic-style information extraction (see placeholders in Figure 2), challenging agents to extract and reason over structured representations, whereas existing benchmarks are limited to extracting fixed, predefined information from web pages. Moreover, the curated databaseInstruct2DS serves as **a valuable resource** for the broader research community, enabling a wide range of downstream applications, including academic recommendation systems, time-series forecasting in finance, game score prediction in sports analytics, and beyond.

## 5 Experiments

To comprehensively evaluate the performance of AutoData, we conduct experiments over Our newly collected dataset `Instruct2DS`, IE benchmark datasets SWDE, EXTENDED SWDE, and Coding benchmark `HumanEval`. Implementation details about environments, evaluation metrics, and experiment setup are provided in Appendix D.1, D.2, and D.4−D.7, respectively. Additional experiments with LLM variants and error analysis are reported in Appendix E.1 and E.2, respectively.

## 5.1 Experiments over Dataset Instruct2DS

We report the performance of AutoData and baseline methods over dataset Instruct2DS in Table 1. The bolded numbers are the best results and the underlined numbers indicate the runner-up results. We choose existing general MAS methods as baseline methods, including Manus [47], AutoAgent [48], OpenManus [49], and AutoAgents [50]. Besides the general AI agents in the literature, we also report the performance of data collection by human efforts, via Cline [45], and Cursor [46] in Table 1. Detailed discussion about baseline methods is provided in Appendix C.

According to the table, we find that: (i) Manual web data collection exhibits superior performance compared to most general MAS systems, albeit with the increased implementation time, which serves as the primary motivation for our research; (ii) The baseline method, Manus, demonstrates enhanced performance relative to all general MAS approaches, while it requires more time and incurs greater expenses to complete the tasks; (iii) Our method AutoData surpasses all baseline methods in every domain, necessitating less implementation time and incurs lower expenses, underscoring the efficiency and cost efficiency of our proposed approach in comparison to existing MAS methods.

## 5.2 Experiments over IE Benchmark Datasets

To further validate the effectiveness of our proposed method AutoData, we conduct experiments over IE benchmark datasets, i.e., SWDE [15] and EXTENDED SWDE [43, 44], for web crawling tasks and list the result in Table 2. We employ the same evaluation metrics in AutoScraper [53] to assess the performance of our proposed method and baseline methods. Detailed discussion about baseline methods, evaluation metrics, and experiment settings is provided in Appendix C, Appendix D.2, and Appendix D.5, respectively.

According to Table 2, we find that: (i) The baseline method, Manus, demonstrates competitive performance on many metrics. However, its results still leave room for improvement on these benchmark datasets, highlighting the benefits of developing a system specifically tailored for web crawling tasks. (ii) Our proposed method AutoData outperforms baseline methods across most evaluation criteria, demonstrating its effectiveness in web crawling tasks.

## 5.3 Experiments over Coding Benchmark Dataset

As our method contains a group of coding agents to develop web data collection programs, we conduct another experiment over the coding benchmark dataset HUMANEVAL [55]. The experiment settings are provided in Appendix D.6 and the performance is listed in Table 3. According to the table, we observe that: (i) Although AutoData is not specifically designed for code generation tasks, its performance is comparable to that of the baseline method

Table 3: Performance comparison with baseline methods over dataset HUMANEVAL.

| Model | Pass@1 | Model | Pass@1 |
|---|---|---|---|
| PaLM [54] | 36.0 | Codex-175B [55] | 47.0 |
| Self-Col [27] | 74.4 | MetaGPT [26] | 85.9 |
| GPT-4 [56] | 67.0 | GPT-4o [57] | 90.2 |
| Ours + GPT-4 | 86.9 | Ours + GPT-4o | **92.5** |

MetaGPT, through the LLM backbone GPT-4. (ii) Our proposed method AutoData, paired with GPT-4, significantly outperforms GPT-4 and, when using GPT-4o as the backbone, it also shows incremental improvements over GPT-4o, underscoring its effectiveness in programming.

## 5.4 Ablation Study

We conduct an ablation study to analyze the contributions of agents and components in OHCache, as illustrated in Figure 3.

**Effectiveness of Agents.** As removing certain agents will result in unworkable codes, we keep the fundamental agents, i.e., MGR $v_{mgr}$, web agent $v_{web}$, and engineering agent $v_{engr}$, and remove the rest of agents by squads, i.e., research squad

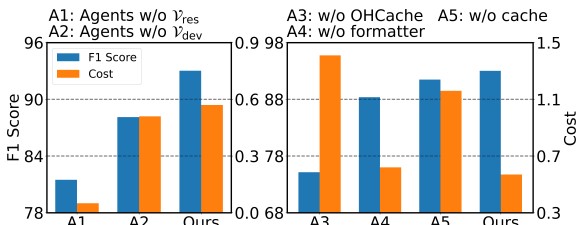

Figure 3: Ablation studies for AutoData.

$\mathcal{V}_{\text{res}}$(A1) and development squad $\mathcal{V}_{\text{dev}}$ (A2), separately. According to the Figure 3, although lower cost, all the ablation settings perform worse than our original setting, indicating that each group of agents contributes to our multi-agent system to some extent.

**Effective of OHCache.** We further validate the effectiveness of each component in OHCache, i.e., the entire OHCache (A3), the oriented hyperedge formatter (A4), and the local cache system (A5), by removing it separately. First, we remove the entire OHCache (A3), which the system switches to broadcast-style communications, omits message formatting, and disables the local cache system. The notable decline in performance and the increase in cost confirm the effectiveness and cost efficiency of OHCache. Next, we remove the oriented hyperedge formatter (A4), which means that we employ a plain natural language for communication, and the performance decline indicates the effectiveness of the oriented hyperedge formatter. Then, we remove the local cache system (A5), which means that the artifacts are embedded in communication. Although performance changes are marginal, substantial cost reduction demonstrates the cost efficiency of the local cache system in our system.

## 5.5 Case Study

To validate the applicability of our method AutoData, we conduct two web data collection tasks for case studies, i.e., children picture book collection and paper collection from surveys. Compared to the tasks in Instruct2DS, these two tasks are more challenging, involving multi-level, in-depth HTML crawling and separate crawling of individual sources.

**Children Picture Book Collection.** We collaborate with researchers in the education domain for children's picture book collections. Picture books are essential for young learner development, and datasets for child picture books hold significant value for contributors in the education fields [58, 59]. See Appendix F for more discussion. Based on the request from domain experts, we choose the Children's Pic-

Table 4: Human evaluation for picture book collection task in accuracy (Acc), completeness (Comp), uniqueness (Uniq), and cost.

| Model | Acc. (↑) | Comp. (↑) | Uniq. (↓) | Cost (↓) |
|---|---|---|---|---|
| Manus | 63.93 | 79.76 | 1.51 | 1.86 |
| AutoData | **89.58** | **98.13** | **0.00** | **0.91** |

ture Book Database at Miami University as the web source, which provides a collection of picture book abstracts searchable by topics, concepts, and skills for building content area literacy across all academic subjects [60]. We compare our proposed system AutoData with the best baseline method Manus to collect all picture books provided in the web source. Afterward, we randomly draw 500 samples from each dataset and conduct the human evaluation. The human evaluation setup is provided in Appendix D.3 and the result is provided in Table 4.

According to the table, we find that (i) Our method consistently outperforms Manus in all human evaluation aspects for the picture book collection task, underscoring the effectiveness of our approach; (ii) AutoData demonstrates significantly lower data collection costs compared to Manus, highlighting its cost-efficiency; (iii) Specifically, AutoData incurs a cost of only \$0.91, half that of Manus, emphasizing both its applicability and cost advantage. Mention that **the collected picture book dataset is shared** with researchers in the education domain for further utilization.

**Paper Collection from Survey.** We conduct another case study to crawl BibTeX formatted references from papers in ArXiv (Experimental HTML). To make the study more challenging and account for the fact that surveys often contain more references than regular papers, we selected five surveys in the MAS field [61–65]. Crawling

Table 5: Performance for paper collection from survey.

| Model | F1. (↑) | Prec. (↑) | Reca. (↑) | Cost (↓) |
|---|---|---|---|---|
| Manus | 74.70 | 87.96 | 61.52 | 2.55 |
| AutoData | **91.16** | **94.28** | **88.46** | **1.40** |

BibTeX from surveys is highly valuable for researchers as it offers an extensive, structured collection of citations that can pinpoint key works and emerging trends within the field. The experiment settings and evaluation metrics are provided in Appendix D.7, and the result is listed in Table 5.

As shown in Table 5, (i) AutoData consistently surpasses Manus across all evaluation metrics, underscoring the superior effectiveness and practical applicability of our approach. (ii) Our method incurs

a lower overall cost compared to Manus, highlighting its enhanced cost-efficiency in challenging web data collection tasks.

# 6 Related Works

**Autonomous Agents for Web Applications.** Website browsers have been one of the major driving forces of LLM agents, as the web sources include all the available applications with up-to-date, domain-specific, and diverse resources of information. Some of the best-known commercial web applications include AI co-scientist [66] and Manus [47]. At the similar times, open-sourced alternatives [49, 48, 67] have been proposed as counter-competitors to speed up the development of this field. However, most existing web applications are created based on general purposes, thus ignoring the specific contexts and challenges of open web data collection tasks, such as the demands of token-intensive operations. As a consequence, these LLM-based web agents have visible performance gaps and redundant LLM calls. Inspired by this, the goal of this work is to develop an LLM-based multi-agent system to automatically collect datasets from open web sources effectively and efficiently.

**Web Information Extraction.** Most websites have semi-structured layouts, e.g., HTML layout, and require models with strong generalizability to understand structural and semantic contexts to extract accurate results [68]. Traditional methods mainly formulate this problem as span-based information extraction tasks [69], such as named entity recognition [70] and entity linking [71], and solve it with sequential labeling [72]. These methods are mainly fine-tuned on pre-trained language models (PLMs). Therefore, high-quality annotated labels or weakly supervised labels are required, which limits its ability to automatically understand diverse layouts of different websites. Recent work [53] leverages generative LLMs to extract web information with in-context learning or prompt tuning. However, these methods require reading every webpage for information extraction, which is expensive and inapplicable for large-scale data collection. In light of this, we propose a multi-agent system that first extracts knowledge for building data collection programs from web sources and further employs development agents to build the scripts for scalable open web data collection.

# 7 Conclusion

In this study, we present a novel multi-agent system (MAS) called AutoData for fully automatic open web data collection. It consists of eight specialized agents that are divided into research and development squads for effective and efficient open web data collection. To tackle the limitations of existing MAS methods in multi-agent collaboration for open data collection tasks, we introduce a novel oriented hypergraph cache system OHCache that includes an oriented hypergraph to depict the complex message sharing process among agents, an oriented hyperedge formatter to structure the messages based on agent profiles, and a local cache system to store the necessary artifacts for agents to access in an on-demand manner. Extensive experiments over a newly collected dataset Instruct2DS, and three existing benchmark datasets demonstrate the effectiveness, efficiency, and applicability of our proposed method, compared with SOTA baseline methods.

## Acknowledgements

The work was partially supported by the NSF under grants IIS-2533550, IIS-2321504, IIS-2340346, IIS-2217239, IIS-2528540, CNS-2426514, and CMMI-2146076, ND-IBM Tech Ethics Lab Program, Notre Dame Strategic Framework Research Grant (2025), and Notre Dame Poverty Research Package (2025). Any expressed opinions, findings, and conclusions or recommendations are those of the authors and do not necessarily reflect the views of the sponsors.

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

# Appendix: Table of Contents

# A  New Dataset Instruct2DS

## A.1  Overview

To comprehensively evaluate the performance of AI agents for data collection from open web sources, we build a new **Instruct**ion to **D**ata**S**et benchmark, called Instruct2DS. The goal of this benchmark dataset is to access the AI agent performance in accomplishing complex and multi-steps tasks in web data collection. Benchmark dataset Instruct2DS contains comprehensive data from various domains, which we will discuss in the following sections. All data in the benchmark dataset Instruct2DS is collected manually, and cross-validated by researchers to ensure the data completeness, quality, and integrity. These data serve as an anchor database for ground truth dataset construction based on the instruction `Instruct`. Given `Instruct` as input, AI agents follow the instruction to collect dataset `DS` from web sources. Besides, our benchmark dataset constructs `GT-DS`, according to the `Instruct`, and we access the performance of the model by comparing the collected dataset `DS` with the ground truth dataset `GT-DS`. Mention that, all models merely take `Instruct` as input, and have **no access** to the data in Instruct2DS in the collection process. The constructed ground truth dataset `GT-DS` is merely used for evaluation. An example of `Instruct` and `GT-DS` is shown in Figure 4.

---

**Task Instruction**

Collect all accepted papers in NeurIPS 2017 Main Conference Track.

---

**A Sample in Ground Truth Dataset**

**TITLE:** Attention is All you Need
**AUTHORS:** Ashish Vaswani, Noam Shazeer, Niki Parmar, Jakob Uszkoreit, Llion Jones, Aidan N Gomez, Łukasz Kaiser, Illia Polosukhin
**ABSTRACT:** The dominant sequence transduction models are based on complex recurrent orconvolutional neural networks in an encoder and decoder configuration. The best performing such models also connect the encoder and decoder through an attentionm echanisms. We propose a novel, simple network architecture based solely onan attention mechanism, dispensing with recurrence and convolutions entirely. Experiments on two machine translation tasks show these models to be superiorin quality while being more parallelizable and requiring significantly less timeto train. Our single model with 165 million parameters, achieves 27.5 BLEU onEnglish-to-German translation, improving over the existing best ensemble result by over 1 BLEU. On English-to-French translation, we outperform the previoussingle state-of-the-art with model by 0.7 BLEU, achieving a BLEU score of 41.1.
**CONFERENCE:** Advances in Neural Information Processing Systems 30 (NIPS 2017)
**ABBR:** NeurIPS
**TRACK:** Main Conference Track
**PAPER_LINK:** Paper Link
**BIBTEX_LINK:** Bibtex Link
**SUPP_LINK**: None

---

Figure 4: Example of instruction `Instruct` and a sample in corresponding `GT-DS`.

There are several concerns in benchmark dataset construction:

- Static of Web Sources: Due to the continuous evolution of web content, ground truth data may change over time, potentially undermining the reproducibility of experimental results. For example, on social media platforms, posts can be edited or deleted, leading to inconsistencies in the collected ground truth across different periods.

- Ethical and privacy concern: Data collection from the open web must adhere to ethical guidelines and respect user privacy. Sensitive or personally identifiable information should be excluded to prevent ethical violations and ensure compliance with relevant data protection regulations.

Based on the aforementioned concerns, we choose three domains that provides static web sources to ensure reproducibility, i.e., academic, finance, and sports. Next, we will discuss each domain in detail.

## A.2 Academic

We choose nine conferences in three fields, i.e., machine learning (NeurIPS, ICLR, and ICML), natural language processing (ACL, EMNLP, and NAACL), and computer vision (CVPR, ICCV, and ECCV), and collect all paper data in these conferences, following the data structure provided in Figure 5. To systematically evaluate the performance of models for data collection tasks in academic papers, we define three instruction templates $\mathbf{T}_{\{1,2,3\}}$ to build instructions [Instruct], listed in Figure 5. With these instruction templates, we implement a ground truth dataset construction algorithm to obtain the ground truth dataset [GT-DS] from Instruct2DS that meets instruction requirements, i.e., [Conference], [Year], [Track], [Start Year] and [End Year].

---

**Academic Paper Data Structure**

**TITLE:** # Paper title.
**AUTHORS:** # Author names, separated by comma.
**ABSTRACT:** # Paper abstract.
**CONFERENCE:** # Full conference name.
**ABBR:** # Abbr of the conference name, i.e., NeurIPS.
**TRACK:** # Conference track of the paper.
**PAPER_LINK:** # The permanent paper link provided by conference.
**BIBTEX_LINK:** # The permanent bibtex link provided by conference.
**SUPP_LINK:** # The permanent supplemental source link provided by conference.

---

**Instruction Templates for Academic Paper Data Collection**

$\mathbf{T}_1$. Collect all papers accepted in [Conference] [Year].
$\mathbf{T}_2$. Collect all papers accepted in [Conference] [Year] [Track].
$\mathbf{T}_3$. Collect all papers accepted in [Conference] from [Start Year] to [End Year].

---

Figure 5: Academic paper data structure and instruction templates.

## A.3 Finance

In addition to academic papers, our benchmark dataset incorporates financial market data from the S&P 500 index. The dataset spans a comprehensive time period from January 1st, 2015, to December 31st, 2024, capturing daily trading information for all S&P 500 constituent companies [73]. The stock dataset includes both standard OHLCV (Open, High, Low, Close, Volume) data [74] and corporate action-adjusted attributes, including adjusted opening price (adjOpen), adjusted daily high (adjHigh), adjusted daily low (adjLow), adjusted closing price (adjClose), adjusted trading volume (adjVolume) and cash dividends per share (divCash). The complete data structure and instruction templates are presented in Figure 6. Our ground truth dataset construction algorithm follows the instruction [Instruct] to obtain the ground truth dataset [GT-DS] that meets the requirements, including [Stock], [Year], [Start Date], [End Date], [Stocks], and [Date].

## A.4 Sports

Sport is another data source that meets our requirements. We build benchmark dataset among two popular sport league, including National Baseketball Association (NBA) and Major League Baseball (MLB).

**NBA.** Our benchmark NBA dataset is sourced from the official NBA website.[1] The dataset encompasses comprehensive statistics from 1946 to 2024, covering two distinct levels: team and player. For

---

www.NBA.com


Stock Market Data Structure - Team Level

**Date**: # The date in format YYYY-MM-DD.
**Open**: # Open trading day price.
**High**: # Highest price within the trading day.
**Low**: # Lowest price within the trading day.
**Close**: # Close trading day price.
**Volume**: # Total number of shares traded within the trading day.
**adjOpen**: # Open trading day price, adjusted by coporate actions.
**adjHigh**: # Highest price within the trading day, adjusted by coporate actions.
**adLow**: # Lowest price within the trading day, adjusted by coporate actions.
**adjClose**: # Close trading day price, adjusted by coporate actions.
**adjVolume**: # Total number of shares traded within the trading day, adjusted for stock splits.
**divCash**: # The cash dividend distrbuted per share.



Instruction Templates for Stock Market Data Collection

$T_4$. Collect daily stock information for [Stock] in [Year].
$T_5$. Collect daily stock information for [Stock] between [Start Date] and [End Date].
$T_6$. Collect daily stock information for [Stocks] on [Date].


Figure 6: Stock data structure and instruction templates.

consistency and comparability, we focus exclusively on regular season data, excluding pre-season and playoff games, as not all teams participate in these additional competitions. The data structure for team and player levels, and corresponding instruction templates are provided in Figure 7 and Figure 8, respectively.

**MLB.** Similarly, our dataset is obtained from the official MLB website. The dataset encompasses comprehensive statistics from 1876 to 2024, covering team-level statistics from two aspects: hitting and pitching. Again, we merely focus on regular-season data. The data structure for hitting and pitching and corresponding instruction templates are provided in Figure 9 and Figure 10, respectively.

## B  Benchmark Datasets

**SWDE.** [15] SWDE is a large, real-world collection of web pages created to support research on the automatic extraction of structured data-such as attribute-value pairs of entities-from the web. It contains webpages from 80 websites in 8 domains, with 124,291 webpages. Each of the websites from the same domains focuses on 3-5 attributes in the web pages.
**Extended SWDE.** [43, 44] This extension provides additional annotations for 21 websites across 3 domains, labeling all binary relations involving the main entity on each page, rather than just a limited set of attributes as in the original SWDE. The new benchmark includes both the extracted object values and the predicate strings, offering a richer and more comprehensive ground truth for evaluating extraction systems.
**HumanEval.** [55] The HumanEval dataset is a benchmark developed by OpenAI to evaluate the code generation capabilities of LLMs. It consists of 164 hand-crafted problems and has become a standard reference for assessing how well AI models can understand, reason about, and generate correct code solutions for programming tasks.

## C  Baseline Methods

To evaluate the effectiveness of our proposed methods, we compare AutoData with baseline methods in the literature and industry, which can be categorized into fthe ollowing groups.

---

https://www.mlb.com/

Figure 7: NBA data structure at the team level and instruction templates.

**Genetic Multi-Agent Systems.** We choose four SOTA multi-agent systems in industry and literature as baseline methods.

- **Manus** [47]. Manus is an AI agent designed to assist with a wide range of tasks, including information gathering, coding, process automation, and general problem-solving using computer and internet resources. Given the high costs of Manus, we conducted our experiments on a representative subset of tasks to ensure cost-effectiveness while maintaining experimental validity. Manus employs a proprietary credit-based pricing system rather than conventional token-based billing. In accordance with their pricing structure, we quantify the expenses using credits, with each credit unit valued at $0.01.

- **AutoAgent.** [48] It introduces a natural language-driven system that enables non-technical users to automatically create, customize, and deploy LLM agents through conversational commands, eliminating programming requirements via its modular architecture and self-managing workflows.

- **OpenManus.** [49] It is an open-source general multi-agent system to solve complex tasks. We obtain their source code from Github.

- **AutoAgents.** [50] It is a framework that dynamically generates, coordinates, and refines multiple specialized agents with distinct roles to form collaborative AI teams tailored to specific tasks, enabling adaptive and coherent multi-agent problem-solving.

**Programming Assistants and Human Programming.** Conventional methods for data collection involve programming to efficiently collect raw data from web sources and clean up the raw data to a ready-to-use dataset [8, 9]. We employ existing programming assistant methods in the industry for data collection, which include:

- **Cursor.** [46] An AI-empowered code editor that acts as a coding assistant, helping developers to work efficiently.

┌─────────────────────────────────────────────────────────────────────┐
│ **NBA Data Structure - Player Level**                                 │
├─────────────────────────────────────────────────────────────────────┤

**PLAYER_NAME**: # The full name of the player.
**YEAR**: # The year which the statistics are recorded.
**BIRTHDATE**: # The birthdate of the player.
**COUNTRY**: # The country of origin.
**HEIGHT**: # The height of the player (feet-inch).
**WEIGHT**: # The weight of the player (pounds).
**DRAFT_YEAR**: # The year the player was drafted into the NBA.
**DRAFT_ROUND**: # The round in which the player was drafted.
**DRAFT_NUMBER**: # The overall draft pick number for the player.
**TEAM_ABBR**: # The Abbr Team Name player associated with during YEAR.
**GP**: # Games Played.
**GS**: # Games Started - Number of games the player started.
**MIN**: # Total Played Minutes.
**FGM**: # Field Goals Made.
**FGA**: # Field Goals Attempted.
**FG3M**: # Three-Point Field Goals Made.
**FG3A**: # Three-Point Field Goals Attempted.
**FTM**: # Free Throws Made.
**FTA**: # Free Throws Attempted.
**OREB**: # Offensive Rebounds.
**DREB**: # Defensive Rebounds.
**REB**: # Total Rebounds.
**AST**: # Assists.
**STL**: # Steals.
**BLK**: # Blocks.
**TOV**: # Turnovers.
**PF**: # Personal Fouls.
**PTS**: # Points.

Figure 8: NBA data structure at player level and instruction template.

- **Cline.** [45] Cline is an open-source AI coding assistant (Copilot) with dual Plan/Act modes, terminal execution, and Model Context Protocol (MCP) for VS Code.

- **Human.** Besides programming assistants in industry, we also conduct data collection via human efforts as a baseline method. Specifically, we require three experts in web crawling to manually program data collection scripts to obtain the datasets DS based on the instructions Instruct. Similar to Manus, we leverage a representative subset of tasks to ensure labor effectiveness while maintaining experimental validity.

For each data collection task, we feed the instructions to the programming assistant to generate data collection Python scripts. Afterward, we run the code for data collection as output. Note that since these methods typically lack direct access to web sources for coding, besides the dataset instructions, we also supply programming assistants with guidelines, including target data sources or preferred collection methods, such as web crawling or REST APIs.

**Web Crawler.** We also compare our method with three baseline methods over IE benchmark datasets, including SWDE [15] and EXTENDED SWDE [43, 44].

---

https://code.visualstudio.com/

## MLB Data Structure - Hitting

**TEAM_NAME**: # The full name of the team.
**TEAM_ABBR**: # The abbreviation of the team.
**YEAR**: # The calendar year of the season.
**AB**: # The number of official at-bats.
**R**: # The number of runs scored.
**HR**: # The number of home runs.
**H**: # The number of hits reaching at least first base.
**2B**: # The number of doubles (reaching second base).
**3B**: # The number of triples (reaching third base).
**RBI**: # The number of runs batted in.
**BB**: # The number of walks (bases on balls).
**SO**: # The number of strikeouts.
**SB**: # The number of stolen bases.
**CS**: # The number of times caught stealing.

## Instruction Templates for MLB Hitting Data Collection

$T_{13}$. Collect MLB [Team] all regular season hitting stats until 2024.
$T_{14}$. Collect MLB [Team] regular season hitting stats from [Start Year] to [End Year].
$T_{15}$. Collect all MLB teams regular season hitting stats in [Year].

Figure 9: MLB hitting data structure and instruction templates.

## MLB Data Structure - Pitching

**TEAM_NAME**: # The full name of the team.
**TEAM_ABBR**: # The abbreviation of the team.
**YEAR**: # The calendar year of the season.
**W**: # The number of wins.
**L**: # The number of losses.
**ERA**: # The earned run average (runs allowed per nine innings).
**G**: # The number of games pitched.
**GS**: # The number of games started.
**CG**: # The number of complete games pitched.
**SHO**: # The number of shutouts pitched.
**SV**: # The number of saves.
**IP**: # The number of innings pitched.
**H**: # The number of hits allowed.
**R**: # The number of runs allowed.
**ER**: # The number of earned runs allowed.
**HR**: # The number of home runs allowed.
**HB**: # The number of hit batters.
**BB**: # The number of walks allowed (bases on balls).
**SO**: # The number of strikeouts.

## Instruction Templates for MLB Pitching Data Collection

$T_{16}$. Collect MLB [Team] all regular season pitching stats until 2024.
$T_{17}$. Collect MLB [Team] regular season pitch stats from [Start Year] to [End Year].
$T_{18}$. Collect all MLB teams' regular season pitching stats in [Year].

Figure 10: MLB pitching data structure and instruction templates.

- **AutoScraper.** [53] An SOTA LLM-powered framework that generates adaptable web scrapers through hierarchical HTML analysis and progressive learning.
- **COT.** [51] This work introduces chain-of-thought prompting for LLMs to decompose the problems into intermediate reasoning steps to improve the performance of LLMs over complex tasks.
- **Reflexion.** [52] It is a framework that enables language agents to improve their decision-making and task performance by generating and storing verbal self-reflections based on feedback from their actions, allowing them to iteratively learn from experience without updating model weights.

# D    Implementation Details

## D.1    Environments

All experiments are conducted on Linux servers equipped with four Nvidia A40 GPUs. The models are implemented using PyTorch 2.4.0 with CUDA 12.1 and Python 3.11.5. We choose GPT-4o [56] as our default LLM backbone for baseline methods and AutoData.

## D.2    Evaluation Metrics

As our focus is on textual data collection, we utilize standard evaluation metrics commonly employed in information extraction tasks, i.e., F1 score, precision, and recall. Furthermore, we also report the time needed to finish the tasks. Mention that, we do not consider the time in program execution, as it is highly uncontrollable due to the several facts, e.g., web sources, Rest API call limits, etc. Instead, we focus on the time that involve human efforts, which is the period from the task instruction release to start data collection. For IE benchmark datasets, i.e., SWDE [15] and EXTENDED SWDE [43, 44], besides the standard evaluation metrics, we also employ execution evaluation metrics [53], which include correct, execution precision (Exec-Prec.), and execution recall (Exec-Reca.). We treat each attribute within the data samples as an individual unit, evaluating and reporting the mean scores across the ground truth dataset $\boxed{\text{GT-DS}}$ and collected data $\boxed{\text{DS}}$. To maintain consistency in numerical comparisons, we round all numerical attributes to four decimal places.

## D.3    Human Evaluation Setup for Case Study

We collaborate with researchers in the education domain for case study children's picture book collections. With the collected dataset, we randomly draw 500 samples and conduct a comprehensive human evaluation based on the following aspects:

- **Completeness**: Whether the sample contains all the attributes provided in the web source.
- **Accuracy**: Whether the value of each attribute correctly reflects the corresponding information on the web source.
- **Uniqueness**: Whether there is any unwanted duplication in the attribute values, ensuring that each attribute represents unique information.

We regard each aspect as a binary classification task, and three researchers evaluate the result separately. Afterward, we conduct a majority vote among three researchers as the final annotation result.

## D.4    Experiment Setup for Dataset Instruct2DS

To ensure comprehensive evaluation across different domains, we sample valid tasks from each instruction template, where a valid task is defined as one that yields non-empty results containing the necessary data samples specified in the instruction. Specifically, for academic paper collection, we generated 3 distinct tasks per conference, resulting in a total of 81 academic paper collection tasks. Similarly, for stock market data collection, we created 27 unique tasks per template, leading 81 stock market collection tasks. In the sports domain, we sampled 6 valid tasks for each instruction template, producing 72 sports-related data collection tasks. This sampling strategy resulted in a diverse evaluation set comprising 234 unique tasks spanning three distinct domains within our dataset Instruct2DS.

### D.5 Experiment Setup for Benchmark Datasets SWDE and EXTENDED SWDE

For experiments over IE benchmark datasets, i.e., SWDE [15] and EXTENDED SWDE [43, 44], we follow the setup in AutoScraper [53] that chooses three seed webpages for models to identify knowledge for web crawler programming, and the rest for testing. We strictly follow the source code of AutoScraper to reproduce the experiment result. Furthermore, to ensure a fair comparison with AutoScraper, we also employ GPT4-Turbo in our experiments for all methods, with the exception of Manus, where the choice of backbone LLMs is not available.

### D.6 Experiment Setup for Benchmark Dataset HUMANEVAL

We follow the settings in MetaGPT [26] to reproduce the experiment results. For AutoData, we simply pass the coding task instruction to the manager agent and request it to manage the workflow among research and development squads for code generation. Moreover, similar to MetaGPT, we slightly modify the prompt to align with the format requirements in the dataset to address format-specific issues, i.e., Python problems.

### D.7 Experiment Setup for Case Study Paper Collection from Survey

For the survey-paper case study, we reuse the precision, recall, and F1 metrics defined in Appendix D.2. Due to the absent of authoritative reference set, two researchers independently extracted every BibTeX entry cited in the target surveys, then cross-validate to ensure the correctness of the BibTex. We use the manually collected BibTex as a ground truth dataset GT-DS to evaluate the collected dataset DS via AutoData and the baseline method Manus.

## E  Additional Experiments

### E.1 Performance of AutoData with different LLMs

To further validate the performance of AutoData with different LLM backbones, we randomly sample one task description from each unique instruction template in dataset Instruct2DS, and evaluate the performance of LLM backbones, i.e., GPT-4o, GPT-4o-mini, DeepSeek R1 70B, and DeepSeek R1 450B. Besides, for better comparison, we also provide the performance of the best baseline method, Manus in Table 6.

According to the table, we find that: (i) For simple tasks, such as STOCK and SPORT, all LLM variants perform similarly, but GPT-4o mini achieves faster execution times and lower costs. (ii) The open-source LLM, DeepSeek R1 70B, shows the lowest performance among all LLMs; however, it still performs comparably to the best baseline method, Manus. (iii) GPT-4o offers the best overall performance in exchange for a relatively high expense among all LLMs.

Table 6: The performance of AutoData with different LLMs backbone over Instruct2DS.

| Model | ACADEMIC | | | STOCK | | | SPORT | | | Time (↓) | Expense (↓) |
|---|---|---|---|---|---|---|---|---|---|---|---|
| | F1 | Prec. | Reca. | F1 | Prec. | Reca. | F1 | Prec. | Reca. | Mins | USD ($) |
| **Manus** | 69.27 | 72.76 | 66.10 | 95.24 | 97.10 | 93.47 | 87.48 | 93.12 | 81.80 | 15.37 | 2.49 |
| **DeepSeek R1 70B** | 78.33 | 85.45 | 72.43 | 94.64 | 96.90 | 91.83 | 85.75 | 90.94 | 80.63 | 13.30 | – |
| **DeepSeek R1 450B** | 90.20 | 92.52 | 88.19 | 95.74 | **98.46** | 93.26 | 87.70 | 94.27 | 80.36 | 4.45 | 0.14 |
| **GPT-4o-mini** | 86.97 | 90.32 | 83.10 | 95.84 | 97.36 | 93.94 | 88.18 | 93.97 | 83.40 | **3.71** | **0.01** |
| **GPT-4o** | **91.85** | **93.74** | **90.51** | **96.75** | 98.37 | **94.17** | **90.14** | **95.63** | **85.28** | 5.58 | 0.57 |

### E.2 Error Analysis

We perform an analysis by looking at the collected datasets DS via AutoData with ground truth datasets GT-DS, and identify the following common failure modes.

**Unclear Conference Name and Track.** Some conferences do not expose the conference name or track on individual paper pages. For instance, in NLP venues, the page only lists a "Volume" string, e.g., "Proceedings of the 62nd Annual Meeting of the Association for Computational Linguistics

(Volume 1: Long Papers)", and the track information is implicitly encoded inside that field. In these cases AutoData misses or mis-parses the attributes. More task-specific prompts could help, but as our goal is a general-purpose MAS for open web data collection, we do not hard-code such heuristics.

**Data Collection Approaches.** Although the system can both crawl HTML and call REST APIs, it occasionally chooses the less reliable route. For MLB statistics, both (i) direct HTML scraping from URL and (ii) the well-maintained third-party APIs are viable. AutoData sometimes opts for crawling, which is more error-prone, instead of the cleaner API path. Better decision policies or explicit user hints may alleviate this issue.

## F  Additional Discussion for Children Picture Book Collection

Picture books, with their creative imagery and engaging narratives, help young learners connect reading with visualization, allowing them to make sense of text through images. Picture books can also serve as powerful tools for students to explore the world that they live in [58]. Moreover, an earlier study [59] discusses how picture books can function as windows, mirrors, and sliding doors, – providing glimpses into the lives of others, reflecting readers' own identities, and opening pathways to new perspectives.

The dataset holds significant value for educators, authors, illustrators, and publishers, serving as a powerful resource for guiding content selection and shaping future publications. Developing a database to identify picture books aligned with specific teaching themes offers an essential tool for analyzing recurring themes in children literature. This not only supports educators in selecting books that promote positive impact in the classroom, but also helps content creators, such as authors, illustrators or publishers, recognize gaps in the field and develop more relevant, representative content.

## G  Limitations

Despite the strong empirical results, AutoData still has several limitations. First, the entire pipeline depends on large-scale proprietary language models (GPT-4 / GPT-4o in our experiments). Any change in pricing, rate limits, or model quality immediately affects latency, reproducibility, and cost, the same for the proprietary baseline method, i.e., Manus. Although our abstraction layer allows the use of open-weight models such as DeepSeek-R1, the superior performance is not guaranteed, as illustrated in Appendix E.1. Second, our benchmark and case studies focus on publicly accessible HTML pages or via REST APIs. Websites rendered entirely client-side, protected by login walls, CAPTCHAs, or aggressive anti-bot policies, remain largely out of reach. Incorporating a headless-browser agent and credential-management modules is left for future work. Third, the current system assumes that the requested data is legally retrievable. Consequently, there is a risk of unintentionally violating copyright or contractual constraints. Users must be mindful of ethical concerns, such as bias in the data, and ensure responsible use of the AutoData in different applications. Fourth, our proposed system merely focuses on text-centric data collection. Multimodal assets (images, audio, video) sources are not supported, and we leave it as further work.

## H  Broader Impacts

AutoData's ability to turn a single sentence into a curated dataset carries both promising benefits and non-trivial risks. On the positive side, the system lowers the barrier to data acquisition for researchers, educators, journalists, and NGOs that lack extensive engineering support. By automating repetitive scraping and cleaning, it frees human time for higher-level analysis and could accelerate meta-studies or reproducibility audits in scientific fields. Our picture-book case study illustrates how domain experts can quickly build specialised corpora that would otherwise take weeks of manual effort. Economically, the framework may increase efficiency and foster new data-driven products, but it also threatens traditional web-scraping or data-annotation jobs. Carefully designing re-skilling programs will be important if adoption scales. There are, however, serious dual-use concerns. Uncontrolled harvesting could capture personal data, pay-walled content, or copyrighted media, exposing operators and downstream users to legal liability. A determined adversary might exploit AutoData to assemble large disinformation corpora or to feed models that generate convincing deepfakes. The environmental footprint is another worry: multi-agent LLM inference, if run at a web-scale, consumes non-negligible

energy. To mitigate these issues, before we release the code, we will opt-in following safeguards: (i) an extensible module that parses common licenses, issuing warnings or blocking non-compliant requests; (ii) a PII detector that redacts sensitive attributes before storage; (iii) rate-limiting and respect for API quotas; and (iv) auditable provenance logs through OHCache so third parties can verify data origin. We also share only task descriptions and checksums of collected artifacts by default, leaving raw dumps under their original licenses. Ultimately, we hope the community will treat AutoData as a research platform—useful for exploring responsible automation, but requiring continuous refinement of legal, ethical, and technical safeguards before it can be deployed safely at scale.

