# OpenReview forum: "AutoData: A Multi-Agent System for Open Web Data Collection"
_NeurIPS.cc/2025/Conference — NeurIPS 2025 poster_

### Official Review · Reviewer_ehRe · 2025-06-20

**Clarity:** 4
**Significance:** 3
**Originality:** 3
**Rating:** 4
**Confidence:** 4

**Summary:**

The paper presents a multi-agent system for collecting data from the web. The proposed MAS has three components, a Research Squad which handles the job for decomposing the instruction into subgoals and forming plans to extract information from the web; a Development Squad takes the plan and try to write code to execute the plan (including testing and validation); and a Manager agent which orchestrates the entire workflow. The MAS protocol is implemented as an Oriented Message Hypergraph which is claimed to be more efficient in terms of cost and empirically have better performance. Considering the difficulty of evaluating the open web data collection task, the paper also proposes a new benchmark, Instruct2DS. Empirical results on the Instruct2DS benchmark and also additional Information Extraction benchmark show that the proposed AutoData outperformed general Web or Coding agents such as Manus and Cursor.

**Questions:**

1. In Ablation setting 3, when removing the graph structure, how does the agent know how to communicate and who to perform first? What’s exactly the implementation of this ablation here?

**Ethical Concerns:**

["NO or VERY MINOR ethics concerns only"]

**Final Justification:**

I thank the authors for the detailed rebuttal and additional analysis; I think my major questions are addressed; I will keep my already positive rating.

**Limitations:**

The authors did not specifically discussed about limitations; please see weakness for suggestions on improvements

**Quality:**

3

**Strengths And Weaknesses:**

# Strengths
1. The paper is well-written with comprehensive evaluations.
2. The proposed benchmark is a good contribution: especially the setting where we consider collecting the latest data from the web is both realistic and useful. The impact can extend to never-ending learning of recent large foundation models.
3. The empirical results against mainstream web/code agents is promising, especially considering a faster turn around time and lower expense

# Weakness

1. Some important comparison for showing the necessity and effectiveness of the proposed MAS frame seems unclear. Specifically, in Table 1, one important comparison should be comparing previous/baseline agentic frameworks equipped with the same backbone model (i.e., GPT-4o) as the AutoData. It is unclear what backbone models are used for most baselines in the table.
  - Also a very related prior work is not compared with nor mentioned: “INFOGENT: An Agent-Based Framework for Web Information Aggregation” (https://arxiv.org/pdf/2410.19054)

2. I feel there is potentially an overcomplication / a lack of novelty in the Hypergraph and Cache System. If I understand correctly, all nodes and edges are predefined. Thus, simply keeping a shared information pool + each agent takes only partial information from the pool should work? This is not a new idea which has been investigated in multiple previous work, such as “Meta-Prompting:Enhancing Language Models with Task-Agnostic Scaffolding” (https://arxiv.org/pdf/2401.12954) and “Mobile-Agent-E: Self-Evolving Mobile Assistant for Complex Tasks” (https://arxiv.org/pdf/2501.11733)

3. See question 1.

---

> ### Author Rebuttal · Authors · 2025-07-30
>
> We sincerely thank the reviewer for the detailed and thoughtful evaluation of our work. We are pleased to hear that the paper is well-written, our experiments are comprehensive, and the reviewer recognizes the contribution of our new benchmark dataset and the promising empirical results against mainstream web/code agents. In this rebuttal, we aim to answer your questions as follows.
>
> > **Q1: Some important comparison for showing the necessity and effectiveness of the proposed MAS frame seems unclear. Specifically, in Table 1, one important comparison should be comparing previous/baseline agentic frameworks equipped with the same backbone model (i.e., GPT-4o) as the AutoData. It is unclear what backbone models are used for most baselines in the table.**
>
> We appreciate your valuable feedback. We choose GPT-4o as our default LLM backbone for AutoData and **all baseline methods except Manus**. For Manus, we do not have options to choose the backbone model, and we use the default model provided by Manus. Thank you for your suggestion. We will clarify this in our next revision.
>
> > **Q2: Performance comparison with prior work INFOGENT: An Agent-Based Framework for Web Information Aggregation.**
>
> Thank you for pointing out INFOGENT. We compare it with AutoData over our benchmark dataset, Instruct2DS, and report the performance in Table 1.
>
> Table 1. Performance comparison (F1) between INFOGENT and AutoData over Instruct2DS.
> | Model | ACADEMIC | STOCK | SPORT  |
> |---|---|---|---|
> | INFOGENT (API) | --  | 68.39 | 63.40|
> | INFOGENT (Visual) | -- | -- | -- |
> | AutoData | 91.85 | 96.75 | 90.14 |
>
> We compare AutoData with INFOGENT in direct API and interactive visual modes. According to Table 1, INFOGENT in both modes fails on tasks in the academic domain. For tasks in stock and sport domains, AutoData has superior performance to INFOGENT, which again demonstrates the effectiveness of our proposed method in open-web data collection tasks.
>
>
>
> > **Q3: I feel there is potentially an overcomplication / a lack of novelty in the Hypergraph and Cache System. If I understand correctly, all nodes and edges are predefined. Thus, simply keeping a shared information pool + each agent takes only partial information from the pool should work? This is not a new idea which has been investigated in multiple previous work, such as “Meta-Prompting: Enhancing Language Models with Task-Agnostic Scaffolding” and “Mobile-Agent-E: Self-Evolving Mobile Assistant for Complex Tasks”.**
>
>
> Thank you for your valuable comments. We would like to clarify that OHCache serves as a communication architecture rooted in **a novel data structure**, called the Oriented Message Hypergraph, to store and retrieve messages or cache files. OHCache introduces **a new perspective** to view the multi-agent collaboration process as an oriented hypergraph. We believe every component in OHCache **is necessary** to address three critical limitations in MAS for open-web data collection tasks:
>
> **Information overload from broadcast messaging.** Unlike existing works, e.g., AgentPrune [1] and G-Designer [2], that employ pairwise graphs for agent communication, we design a novel data structure called Oriented Message Hypergraph to enable one-to-many message sharing.
>
> **Ineffective unstructured natural language interface.** Existing works, e.g., Meta-Prompting and Mobile-Agent-E, commonly employ natural language for communication, which may be ineffective in message sharing. To handle this challenge, AutoData leverages the oriented hypergraph formatter to structure the messages into JSON format. The structured messages allow subsequent agents to capture salient information for efficient complex problem-solving. Moreover, the oriented hypergraph formatter is responsible for inserting the message into the oriented message hypergraph based on the pre-defined relationship. During development, we have tested agents to select message recipients dynamically. However, this approach **hinders agents from performing their primary tasks** and **increases system complexity without corresponding benefits**. Our current role-based solution already achieves superior performance compared to baseline methods while maintaining simplicity.
>
> **Propagation of unnecessary artifact-embedded messages.** We introduce a local cache system to store artifacts so that all agents can access these artifacts on demand, without polluting the communication channel and preventing the token waste in the propagation of artifact-embedded messages.
>
> The ablation study in Section 5.4 **validates the necessity of each component in our OHCache**. As illustrated in Figure 3, the performance drop and cost increase by removing the entire OHCache, or a partial mechanism, i.e., the formatter and cache system, demonstrate the necessity of each component in OHCache.
>
> We further emphasize the contributions of our work as follows:
> 1. We design a **fully automatic multi-agent system** called AutoData for open web data collection tasks, which consists of eight specialized agents and a novel oriented hypergraph cache system **for efficient and effective multi-agent collaboration.**
> 2. We introduce a newly collected benchmark Instruct2DS, which is **first benchmark dataset** to evaluate the performance of models in open web data collection settings.
>
>
> > **Q4: In Ablation setting 3, when removing the graph structure, how does the agent know how to communicate and who to perform first? What’s exactly the implementation of this ablation here?**
>
>
> The entire workflow is orchestrated by `Manager Agent`, which is responsible for deciding the next agent for execution. For ablation A3, we remove our proposed `OHCache`, and employ broadcast-style communications, same as existing work, e.g., CAMEL [3], etc, where every message is disseminated to all agents and each agent processes the entire message history for decision-making.
>
>
> > **Q5: The authors did not specifically discuss limitations; please see the weakness for suggestions on improvements.**
>
> Due to the limited space in the main paper, we have detailed discussions for limitations and potential social impact in **Appendix G and H**, respectively. Thank you for your suggestions. We will add the above discussion in our next revision.
>
>
> [1] Cut the Crap: An Economical Communication Pipeline for LLM-based Multi-Agent Systems. ICLR’25.
>
> [2] G-Designer: Architecting Multi-agent Communication Topologies via Graph Neural Networks. ICLR’25 Workshop.
>
> [3] CAMEL: Communicative Agents for "Mind" Exploration of Large Language Model Society. NeurIPS’23.

---

> > ### Comment · Reviewer_ehRe · 2025-07-31
> >
> > I thank the authors for the detailed rebuttal and additional analysis; I think my major questions are addressed; I will keep my already positive rating.

---

> > > ### Author Response · Authors · 2025-08-01
> > >
> > > Thanks for your support and constructive comments again! Your review is really helpful for our work!

---

### Official Review · Reviewer_8H7h · 2025-06-27

**Clarity:** 3
**Significance:** 3
**Originality:** 3
**Rating:** 4
**Confidence:** 4

**Summary:**

The research introduces AutoData, a novel multi-agent system designed for automated web data collection that minimizes human input, requiring only a natural language instruction. This system features a robust multi-agent architecture, coordinated by a central task manager, which efficiently organizes agents into research and development squads. A key innovation is the Oriented HyperGraph Cache System (OHCache), which enhances collaboration among agents and significantly reduces token costs commonly associated with existing Large Language Model (LLM)-based systems. To validate its effectiveness, the authors also present Instruct2DS, a new benchmark dataset supporting live data collection from various web sources, demonstrating AutoData's superior performance in terms of efficiency and accuracy compared to current methods.

**Questions:**

1. The paper indicates a dependency on proprietary LLMs like GPT-4 and GPT-4o, noting that the superior performance with open-weight models (such as DeepSeek-R1) is not guaranteed. Given that GPT-4o offers the best overall performance at a relatively high expense, are there specific plans to further optimize AutoData's architecture or conduct targeted fine-tuning on open-weight LLMs to reduce reliance on proprietary models while maintaining competitive performance and improving cost-efficiency for wider adoption?
2.  The current system is limited to text-centric data collection and does not support multimodal assets like images, audio, or video. What is the roadmap for extending AutoData's capabilities to include diverse multimodal data collection, and would this require a significant overhaul of the existing agent architecture or the introduction of new specialized agents?
3. The error analysis points to two key failure modes: mis-parsing attributes when information is implicitly encoded or conference names/tracks are unclear, and sometimes opting for less reliable HTML scraping even when cleaner REST API options are available. How do the authors plan to enhance AutoData's reasoning and decision policies to overcome these issues, particularly in distinguishing optimal data acquisition methods (e.g., API vs. scraping) and handling ambiguous or implicitly structured web content?

**Ethical Concerns:**

["NO or VERY MINOR ethics concerns only", "Major Concern: Data privacy, copyright, and consent"]

**Limitations:**

Yes

**Quality:**

3

**Strengths And Weaknesses:**

Strengths -
1. AutoData features a robust multi-agent architecture with a novel oriented message hypergraph coordinated by a central task manager that efficiently organizes agents across research and development squads. The system is composed of eight specialized agents, strategically organized into two synergistic squads (research and development), coordinated by a central manager agent (MGR).
2. AutoData addresses the significant token costs prevalent in existing Large Language Model (LLM)-based systems through its novel Oriented HyperGraph Cache System (OHCache).
3. The paper introduces Instruct2DS, the first open web data collection benchmark, which addresses a significant gap in evaluating AI agent performance for such tasks. This dataset spans three domains (academic, finance, and sports). Instruct2DS is distinctive because it requires MAS agents to interact with live, dynamic web sources, supports diverse data acquisition modalities (including REST API calls), and enables symbolic-style information extraction.

weaknesses -
1. The current system primarily focuses on publicly accessible HTML pages or REST APIs. It struggles with websites that are entirely client-side rendered, protected by login walls, CAPTCHAs, or aggressive anti-bot policies.
2. The proposed system is currently limited to text-centric data collection and does not support multimodal assets such as images, audio, or video.
3. There is a risk of unintentionally violating copyright or contractual constraints as the system assumes requested data is legally retrievable. Users must be mindful of ethical concerns and responsible use, as uncontrolled harvesting could lead to legal liability.
4. Running multi-agent LLM inference at a web-scale consumes non-negligible energy, raising environmental concerns.

---

> ### Author Rebuttal · Authors · 2025-07-30
>
> We sincerely thank the reviewer for the thoughtful and constructive feedback. We are pleased that the reviewer recognizes the innovative contributions of AutoData and our benchmark contribution of Instruct2DS, i.e., the first open web data collection benchmark. In this rebuttal, we aim to address the reviewer’s concerns as follows.
>
> > **Q1: The current system primarily focuses on publicly accessible HTML pages or REST APIs. It struggles with websites that are entirely client-side rendered, protected by login walls, CAPTCHAs, or aggressive anti-bot policies.**
>
> We acknowledge AutoData's limitations when encountering websites with strong anti-crawling protections. In our submission, to strictly follow the NeurIPS Code of Ethics, we limit our approach to publicly accessible webpages and REST APIs that explicitly permit automated access, prioritizing ethical practices over comprehensive coverage.
>
> If beyond the scope of submission, we are aware that technical solutions exist to collect data among entirely client-side rendered, protected by login walls, CAPTCHAs, or aggressive anti-bot policies. Specifically, to access login-protected content, we can simply extend AutoData with **an authentic agent to interact with the user to request authentication requirements**, e.g., OAuth, form-based login, or API keys, etc. Moreover, this agent would integrate a secure credential storage solution to manage user-provided credentials safely. For CAPTCHA, we advocate an ethical approach that respects website security measures. That is, when AutoData encounters a CAPTCHA, it will pause automation and **request human assistance.** For websites that are client-side rendered or contain aggressive anti-bot policies, we can leverage headless browser solutions, e.g., Playwright, and Selenium, etc, to **render pages in a real browser environment for data collection.**
>
> We would like to clarify that our proposed AutoData currently remains in the scope of **common academic practice**, and should be used only for research or non-commercial purposes. We strictly follow the NeurIPS Code of Ethics and regional legislations, i.e., never attempting to break security measures, only accessing content the user has legitimate rights to access, respecting rate limits, and server resources. We acknowledge that some web sources will remain inaccessible if they have strong anti-automation policies, and this is by design. The objective of Autodata is to automate **legitimate data collection**, not to circumvent security measures.
>
>
> > **Q2: The proposed system is currently limited to text-centric data collection and does not support multimodal assets such as images, audio, or video (originally W2&Q2).**
>
> Thank you for your valuable feedback. As pioneering work in open-source web data collection methodologies, we focus on textual data to establish foundational principles and demonstrate core capabilities.
>
> Extending AutoData to multimedia domains would necessitate substantial architectural modifications, including specialized agents, tools, and multimodal LLM backbones. Specifically, A minimum overhaul would require: (i) **a media agent within the research squad to identify and evaluate multimedia content against user specifications**, (ii) **specialized tools for multimedia processing and storage**, and (iii) **multimodal LLM architectures capable across media and textual modalities.**
>
> Beyond these technical requirements, multimedia data collection introduces several unique challenges that distinguish it from text-based approaches:
>
> **Format Heterogeneity.**  Unlike textual data, which predominantly uses standardized encodings (UTF-8, ASCII), multimedia content exhibits significant format diversity. Images alone encompass various formats, e.g., JPEG, WebP, SVG, AVIF, and GIF, etc. Each requires distinct parsing algorithms, quality assessment metrics, and storage optimization strategies. Furthermore, content delivery networks often serve different formats based on client capabilities (browser type, device specifications, bandwidth), necessitating format detection and adaptive conversion pipelines.
>
> **Deduplication Complexity.** Multimedia deduplication presents substantially greater complexity than text-based approaches. Identical content may appear across multiple resolutions, compression levels, and watermarks, etc. Traditional hash-based deduplication becomes inadequate, requiring more advanced deduplication solutions.
>
> **Legal and Ethical Complexities.** While text-centric collection raises important legal and ethical considerations, media domains exponentially increase these concerns. Media content often involves complex copyright ownership, personality rights, and privacy implications. Notice that several source websites may lack clear licensing information, creating cascading legal uncertainties that are difficult to resolve programmatically.
>
> These challenges represent active areas of research requiring expertise across multiple domains. While we remain optimistic about future solutions, addressing these complexities is beyond the scope of the current work.
>
> > **Q3: There is a risk of unintentionally violating copyright or contractual constraints as the system assumes requested data is legally retrievable. Users must be mindful of ethical concerns and responsible use, as uncontrolled harvesting could lead to legal liability.**
>
> We appreciate the reviewer for pointing out this concern. As discussed previously, AutoData remains in the scope of **common academic practice** and should be used only for **research or non-commercial purposes**. Moreover, AutoData can be easily extended to include (i) **an extensible module that parses common licenses, issuing warnings or blocking non-compliant requests**; (ii) **employ a PII detector to prevent collecting sensitive attributes before storage**; (iii) **auditable provenance logs through OHCache so users can verify data origin**, to tackle the aforementioned challenges. We already discussed these solutions in Appendix H.
>
> > **Q4: Running multi-agent LLM inference at a web-scale consumes non-negligible energy, raising environmental concerns.**
>
> We agree that energy consumption is indeed a concern for web-scale multi-agent systems. AutoData addresses this through two key design choices: (i) the OHCache system minimizes redundant LLM calls through **efficient information sharing** with an oriented message hypergraph and artifact caching. Experiment over Instruct2DS demonstrates that AuoData reduces token usage by 47% compared with OpenManus (lowest cost among baseline methods); and (ii) AutoData generates **executable code for data collection rather than using LLMs** to process each webpage individually, which further reduces energy consumption.
>
>
> > **Q5: The paper indicates a dependency on proprietary LLMs like GPT-4 and GPT-4o, noting that the superior performance with open-weight models (such as DeepSeek-R1) is not guaranteed. Given that GPT-4o offers the best overall performance at a relatively high expense, are there specific plans to further optimize AutoData's architecture or conduct targeted fine-tuning on open-weight LLMs to reduce reliance on proprietary models while maintaining competitive performance and improving cost-efficiency for wider adoption?**
>
> Thank you for this observation. During the development process, we have obtained extensive data consisting of agent-specific input-output pairs that capture the requisite reasoning patterns for web data collection tasks. Open-weight models (e.g., DeepSeek-R1) fine-tuned over our collected gold agent input-output pairs could achieve competitive performance with GPT-4o, addressing the quality disparities observed in our experiments. This represents a promising direction to reduce the reliance on proprietary models and maintain cost-efficient and competitive performance. We regard this direction as our future work.
>
> > **Q6: The error analysis points to two key failure modes: mis-parsing attributes when information is implicitly encoded or conference names/tracks are unclear, and sometimes opting for less reliable HTML scraping even when cleaner REST API options are available. How do the authors plan to enhance AutoData's reasoning and decision policies to overcome these issues, particularly in distinguishing optimal data acquisition methods (e.g., API vs. scraping) and handling ambiguous or implicitly structured web content?**
>
> **Deciding optimal data acquisition methods.**
> AutoData can handle this issue by adding a simple interface agent on top of the current system to interact with users to clarify their preferences for data acquisition methods. This feature is similar to the design in Manus and OpenAI Deep Research. Specifically, interface agent **is only activated** when the user's instruction is unclear or multiple collection methods are available. This agent interacts with the user to clarify unclear instructions and chooses among valid data acquisition methods (The trade-off analysis, e.g., cost, accuracy, and latency, is also provided by the agent). After refining the instruction and clarify data collection method, AutoData **remains automatic for open web data collection** with no human effort. In our experiments, we do not include this agent for **a rigorous and fair comparison** with baseline methods.
>
> **Handling ambiguous or implicitly structured web content.** Intuitively, a more task-specific prompt may alleviate the issue. However, as our goal is to build a general-purpose MAS for open web data collection, we do not hard-code such heuristics. In the long term, as discussed previously, we plan to fine-tune open-weight LLMs, e.g., DeepSeek R1, with our collected agent input-output pairs, which can explicitly tackle these errors.

---

> > ### Comment · Reviewer_8H7h · 2025-08-01
> >
> > Thanks for the response to my concerns. I agree with the explanations and happy with the responses. No further questions from me.

---

> > > ### Author Response · Authors · 2025-08-02
> > >
> > > Thank you again for the valuable and supportive feedback. Your review is really helpful for our work!

---

### Official Review · Reviewer_2gdK · 2025-06-29

**Clarity:** 3
**Significance:** 3
**Originality:** 3
**Rating:** 5
**Confidence:** 4

**Summary:**

This paper introduces AutoData, a fully automated multi-agent system for web-scale data collection from open sources. AutoData organizes agents into two functional squads, research and development. They are coordinated by a manager agent and connected through a novel communication architecture called OHCache (Oriented Hypergraph Cache). OHCache formalizes agent messaging as a directed hypergraph, which enables structured, scoped, and efficient communication. The paper also contributes Instruct2DS, a new benchmark for open web data collection spanning academic, finance, and sports domains. AutoData outperforms both LLM-based and traditional scraper baselines across accuracy, cost, and time.

**Questions:**

- How are role-to-target mappings maintained or adapted? Are they hard-coded or learned over time?
- Could the OHCache framework generalize to other tasks beyond web data collection? Which part of the design would require major changes?

**Ethical Concerns:**

["NO or VERY MINOR ethics concerns only"]

**Final Justification:**

I think the rebuttal addressed my minor concerns about novelty and unstructured user input. Therefore I will maintain my original positive ratings.

**Limitations:**

Yes

**Quality:**

3

**Strengths And Weaknesses:**

Strengths:
- The proposed method encodes inter-agent messaging as oriented hyperedges, which allows targeted message delivery and avoid message broadcasting.
- It introduces token-aware artifact offloading via local cache + cache ID routing, which effectively avoids context bloating
- The proposed method demonstrates reduced cost and execution time in addition to strong empirical performance.

Weaknesses:
- Agent-specific contexts are not uncommon in multi-agent system design, which limits the novelty.
- How the system handles underspecified or vague user instructions and out-of-domain webpages isn’t deeply explored, which raises doubts about its robustness.
- It’s unclear how the manager agent prioritizes conflicting or simultaneous message paths.

---

> ### Author Rebuttal · Authors · 2025-07-31
>
> We thank the reviewer’s valuable and constructive comments. We appreciate the reviewer's recognition of our key contributions, particularly the design of an oriented message hypergraph for efficient inter-agent communication, the artifact caching mechanism that addresses context bloating challenges, and the demonstrated improvements in both computational efficiency and empirical performance. In this rebuttal, we aim to alleviate your concerns as follows.
>
> > **Q1: Agent-specific contexts are not uncommon in multi-agent system design, which limits the novelty.**
>
> We would like to clarify that OHCache serves as a communication architecture rooted in **a novel data structure**, called the Oriented Message Hypergraph, to store and retrieve messages or cache files. Every component in OHCache is necessary to address three critical limitations in MAS for open-web data collection tasks: (i) **information overload from broadcast messaging**, (ii) **ineffective unstructured natural language interface**, and (iii) **propagation of unnecessary artifact-embedded messages**.
>
> Specifically, to tackle the information overload issue, unlike existing works, e.g., AgentPrune [1] and G-Designer [2], that employ pairwise graphs for agent communication, we design a novel data structure called Oriented Message Hypergraph to enable one-to-many message sharing.
>
> Existing works [3, 4] commonly employ natural language for communication, which is ineffective in message sharing. To handle this challenge, AutoData leverages the oriented hypergraph formatter to structure the messages into JSON. The structured messages allow subsequent agents to capture salient information. Moreover, the oriented hypergraph formatter is responsible for inserting the message into the oriented message hypergraph based on the pre-defined relationship.
>
> To prevent the token waste in the propagation of artifact-embedded messages, we introduce a local cache system to store artifacts so that all agents are able to access cached artifacts on demand, without polluting the communication channel.
>
> We further emphasize the contributions of our work:
>
> 1. We introduce a fully automatic multi-agent system, called AutoData, that consists of eight specialized agents and a novel oriented hypergraph cache system for efficient and effective multi-agent collaboration, that turns a **one-sentence data requirement into a ready-to-use dataset.**
> 2. We introduce a newly collected benchmark,k Instruct2DS, the **first benchmark dataset** to evaluate the performance of models for open web data collection tasks.
>
> > **Q2: How the system handles underspecified or vague user instructions and out-of-domain webpages isn’t deeply explored, which raises doubts about its robustness.**
>
> We appreciate the reviewer's valuable comments. For underspecified or vague user instructions, AutoData can extend with a simple interface agent on top of the current system to interact with users to clarify the underspecified or vague requirements. This feature is similar to the design in Manus and OpenAI Deep Research. Specifically, interface agent **is activated** when the user's instruction is unclear. This agent interacts with the user to clarify unclear requirements. After refining the instruction, AutoData **remains automatic for open web data collection** with no human effort. In our experiments, since Instruct2DS has **well-defined instructions**, and for **a rigorous and fair comparison** with baseline methods, we do not include this agent.
>
> To validate the robustness of AutoData beyond the Instruct2DS benchmark, we have conducted two case studies, i.e., children's picture book collection and paper collection from the survey in Section 5.5, to assess the performance of AutoData in domains not included in the benchmark. These two tasks are **more challenging**, involving multi-level, in-depth HTML crawling and separate crawling of individual sources. We compared it with the industry-level method, i.e., Manus. The results demonstrate AutoData's applicability, cost efficiency, and robustness.
>
> > **Q3: It’s unclear how the manager agent prioritizes conflicting or simultaneous message paths**
>
> AutoData employs a sequential agent workflow, as each agent's output influences subsequent workflow decisions. The manager agent serves as the central orchestrator, i.e., determines the agent to execute next based on previous results and messages stored in OHCache. OHCache functions purely as a message-sharing and caching system, which means message delivery itself does not trigger agent execution. This separation ensures the manager agent maintains full control over the execution flow while OHCache optimizes communication among agents. Thank you for the question. We will clarify the workflow of AutoData in our next revision.
>
> > **Q4: How are role-to-target mappings maintained or adapted? Are they hard-coded or learned over time?**
>
> Our role-to-target mappings are simply hard-coded. During development, we have explored agents to dynamically select message recipients. However, this approach hinders agents from performing their primary tasks and increases system complexity without corresponding benefits. Our current role-based message sharing already achieves superior performance compared to baseline methods while maintaining simplicity. The predefined communication patterns ensure agents remain focused on their specialized functions rather than communication logistics, resulting in both better task performance and lower computational overhead.
>
> > **Q5: Could the OHCache framework generalize to other tasks beyond web data collection? Which part of the design would require major changes?**
>
> OHCache framework **is applicable to domains beyond open web data collection** with minimal adaptation: (i) Defining agent roles and communication patterns; (ii) Specifying message formats for the domain; and (iii) Identifying reusable artifacts to cache.
>
> For instance, in a financial trading system, specialized agents would include a market monitor (tracking prices), a news analyzer (processing events), technical analysts (computing indicators/ analysis information), a risk manager (evaluating exposure), and an execution agent (placing trades). When the news agent identifies breaking information, OHCache creates a directed hyperedge from {news agent} → {technical analyst, risk manager}, ensuring only relevant agents receive the message while excluding others (e.g., execution agent) who don't need raw news. The cache system would store computationally expensive artifacts like historical market data, processed news summaries, and technical indicators, preventing redundant calculations and API calls. This selective message sharing and caching would reduce both latency and computational costs compared to broadcast-based communications.
>
>
> [1] Cut the Crap: An Economical Communication Pipeline for LLM-based Multi-Agent Systems. ICLR’25
>
> [2] G-Designer: Architecting Multi-agent Communication Topologies via Graph Neural Networks. ICLR’25 Workshop.
>
> [3] CAMEL: Communicative Agents for "Mind" Exploration of Large Language Model Society. NeurIPS’23.
>
> [4] Meta-Prompting: Enhancing Language Models with Task-Agnostic Scaffolding. Arxiv.

---

### Official Review · Reviewer_ddG4 · 2025-06-30

**Clarity:** 4
**Significance:** 3
**Originality:** 4
**Rating:** 4
**Confidence:** 3

**Summary:**

This paper proposes AutoData, a novel multi-agent system for automated open web data collection. It introduces a structured communication framework called OHCache to reduce message redundancy and improve collaboration efficiency, and presents Instruct2DS, a new benchmark spanning academic, finance, and sports domains. Experimental results demonstrate that AutoData achieves superior performance in accuracy, cost-efficiency, and execution time compared to strong baselines across multiple datasets and real-world tasks. The authors also explicitly acknowledge ethical and privacy concerns in web data collection and design their benchmark accordingly.

**Questions:**

1. Could you provide more details on the “cost” metric used in the experiments? For example, how much would it approximately cost to collect 1,000 samples using GPT-4? Is the data collection cost sensitive to the nature or structure of different web sources?

2. Could the proposed OHCache framework be generalized to other multi-agent domains beyond data collection, such as tool-using or decision-making systems?

3. What is the failure rate or error distribution of AutoData in real-world web scraping scenarios not represented in the Instruct2DS benchmark?

**Ethical Concerns:**

["NO or VERY MINOR ethics concerns only"]

**Final Justification:**

The authors provided a clear rebuttal that addresses most of my concerns. However, the main paper still lacks sufficient discussion on data quality and coverage, which are critical to the system’s practical utility and reliability. Considering this limitation, I prefer to keep my original score.

**Limitations:**

While the authors briefly acknowledge certain limitations—such as web content volatility and ethical/privacy concerns—in the appendix, these issues are not sufficiently addressed in the main paper. Critical points that deserve more discussion include:
**Societal Impact:** The potential for misuse (e.g., scraping sensitive or copyrighted content) and alignment with ethical scraping norms (e.g., robots.txt, rate limits) is not explored in depth.

The authors are encouraged to integrate a dedicated section in the main paper on limitations and societal impact. Concrete discussion on compliance with data usage policies, safety checks during scraping, and value alignment would improve the paper’s transparency and trustworthiness.

**Paper Formatting Concerns:**

There are no major formatting issues in the paper.

**Quality:**

3

**Strengths And Weaknesses:**

### Strengths
1. Strong Practical Motivation: The paper targets a highly practical and relevant problem—web data collection—which remains a complicated and labor-intensive workload for developers.

2. Effective System Design: The proposed system integrates multiple techniques to address key bottlenecks in multi-agent system (MAS) design, notably in improving both effectiveness and efficiency.

3. New Benchmark (Instruct2DS): The introduction of a live, domain-diverse benchmark is a substantial contribution, helping to fill the gap in realistic evaluation datasets for open web scraping tasks.

### Weaknesses

1. Data Quality and Coverage: The practicality of AutoData may be limited by real-world data diversity and quality issues. This could stem from differences in regional data collection policies, as well as engineering challenges such as deduplication, detoxification, safety filtering, and value alignment.

2. Limited Discussion of Failure Modes: While ethical and volatility-related concerns are briefly acknowledged in the appendix, the main paper lacks an in-depth discussion on how the system handles adversarial, highly dynamic, or JavaScript-heavy web environments.

---

> ### Author Rebuttal · Authors · 2025-07-31
>
> We appreciate the reviewer's thoughtful and valuable feedback. We are glad to see the reviewer recognize novelty of AutoData, our strong practical motivation, effective model design, and the contribution of our newly introduced dataset.
>
> > **Q1: Data Quality and Coverage: The practicality of AutoData may be limited by real-world data diversity and quality issues. This could stem from differences in regional data collection policies, as well as engineering challenges such as deduplication, detoxification, safety filtering, and value alignment.**
>
> We thank the reviewer for raising this question. We acknowledge current limitations in safety filtering, toxicity detection, and value alignment. AutoData benefits from two layers of built-in protection: (1) our workflow begins with search engines that employ "safe search" filtering by default, and (2) our backbone LLM, i.e., GPT-4o, incorporates extensive human alignment training. These safeguards provide a baseline level of content safety and value alignment. Moreover, we want to emphasize that **these challenges represent fundamental obstacles faced by all web crawling systems.**
>
> Besides the built-in protection, AutoData incorporates several mechanisms to address these data quality and diversity challenges. Specifically, our test and validation agent in AutoData can handle the multi-level quality checks, including schema validation, completeness verification, and deduplication. As demonstrated in column Uniqueness Table 4 for the case study, AutoData is free of duplication issues.
>
> We would like to further clarify that AutoData usage should strictly follow the NeurIPS Code of Ethics and the regional data collection policies. While this may hinder AutoData's practicality to some extent, we prioritize ethical and legitimate practices over comprehensive coverage. Moreover, AutoData should be used only for **research or non-commercial purposes**. We hope our method can contribute to not only the **traditional AI/ML community** but also **research domains relevant to trustworthiness.**
>
> > **Q2: Limited Discussion of Failure Modes.**
>
> Thank you for pointing out the question regarding failure modes. As a research paper, we mainly focus on methodology. The solutions to handle failure modes belong to development details, which we omit in the submission. To alleviate your concern, we will provide a detailed discussion as follows.
>
> Regarding adversarial environments, we may extend AutoData with **an authentic agent** to interact with the user for assistance to pass CAPTCHAs, or request authentication requirements, e.g., OAuth, form-based login, API keys, etc. Moreover, this agent would integrate a secure credential storage solution to manage user-provided credentials safely. We can leverage headless browser solutions, e.g., Playwright, Selenium, etc., for highly dynamic or JavaScript-heavy web sources to render pages in a real browser environment for data collection. Again, while technical solutions exist, AutoData usage should strictly align with the NeurIPS Code of Ethics and the legitimate requirements. We will include these discussions in our next revision.
>
> > **Q3:Could you provide more details on the “cost” metric used in the experiments? For example, how much would it approximately cost to collect 1,000 samples using GPT-4? Is the data collection cost sensitive to the nature or structure of different web sources?**
>
> We appreciate your question. The expense column in Table 1 is reported based on the average cost per task. As discussed in Appendix D.1, we choose GPT-4o as our default LLM backbone for AutoData, and all baseline methods, except Manus. We utilize the OpenAI dashboard to monitor the cost. For Manus, it employs a proprietary credit-based pricing system rather than conventional token-based billing. Following their pricing structure, we quantify the expenses using credits, with each credit unit valued at $0.01.
>
> We do not have measurement for the cost to collect 1,000 samples using GPT-4o, as (i) the number of samples is dependent on the tasks; (ii) the cost in our method is mainly on explore crawling logics for program development, and if we have finalized the crawling program for a specific task, the number of samples to collect is independent to the cost. For task-level expense, the average cost is $0.57.
>
> The data collection cost is sensitive to the structure of different web sources. Specifically, the main token usage is reading the webpages to extract crawling logic for programming. For academic papers, the web page is roughly 60K tokens, while for Sport-related tasks, the corresponding webpage is only 18K tokens.
>
> > **Q4: Could the proposed OHCache framework be generalized to other multi-agent domains beyond data collection, such as tool-using or decision-making systems?**
>
> We thank the reviewer for recognizing the broader potential of OHCache. **OHCache framework is applicable to domains beyond open web data collection** with minimal adaptation: (i) Defining agent roles and communication patterns; (ii) Specifying message formats for the domain; and (iii) Identifying reusable artifacts to cache.
>
> For instance, in a financial trading system, specialized agents would include market monitors (tracking prices), news analyzers (processing events), technical analysts (computing indicators and analysis information), risk managers (evaluating exposure), and execution agents (placing trades). When the news agent identifies breaking information, OHCache creates a directed hyperedge from {news agent} → {technical analyst, risk manager}, ensuring only relevant agents receive the message while excluding others, e.g., the execution agent, who don't need raw news. The cache system would store computationally expensive artifacts like historical market data, processed news summaries, and technical indicators, preventing redundant calculations and API calls. This selective message sharing and caching would reduce both latency and computational costs compared to broadcast-based communications.
>
>
> > **Q5: What is the failure rate or error distribution of AutoData in real-world web scraping scenarios not represented in the Instruct2DS benchmark?**
>
> Thank you for the question. To alleviate your concern, we provide the error distribution in our case studies that examine AutoData in real-world web scraping beyond Instruct2DS.
>
> | Method | Parsing Error | Structure Error | Access Error |
> | --- |  --- |--- |--- |
> | Manus | 10.89% | 6.75% | **0%**|
> | AutoData | **6.37%** |  **0.27 %** |  **0%** |
>
> The table above categorizes errors into three types: parsing errors (incorrect attribute values), structure errors (missing attributes), and access errors (caused by rate limits or connection refusal). The results show that AutoData consistently achieves lower error rates than Manus across all evaluation criteria, demonstrating AutoData's superior performance in real-world open web data collection tasks.
>
>
> > **Q6: While the authors briefly acknowledge certain limitations—such as web content volatility and ethical/privacy concerns—in the appendix, these issues are not sufficiently addressed in the main paper. Critical points that deserve more discussion include: Societal Impact: The potential for misuse (e.g., scraping sensitive or copyrighted content) and alignment with ethical scraping norms (e.g., robots.txt, rate limits) is not explored in depth. The authors are encouraged to integrate a dedicated section in the main paper on limitations and societal impact. Concrete discussion on compliance with data usage policies, safety checks during scraping, and value alignment would improve the paper’s transparency and trustworthiness.**
>
> Thank you for your suggestions. Due to the limited space in the main paper, we leave the discussion about limitations and the broader social impact in the appendix. In our next revision, we will extend a detailed discussion in our main paper as follows.
>
> While AutoData offers a significant practical solution to collect data from open web sources, it remains in the scope of **common academic practice**, and should be used only for **research and non-commercial purposes**. We hope AutoData can contribute to not only the traditional AI/ML community but also research domains relevant to trustworthiness.
>
> Moreover, we acknowledge several critical societal and ethical considerations that must be addressed:
>
> **Data Usage Compliance**: our released version incorporates built-in safeguards to respect website policies, including an extensible module in the research squad that parses common licenses, robots.txt files, implementation with respect to rate limits, and adherence to API usage guidelines when available.
>
> **Privacy and Sensitivity Protection**: To ensure protection of privacy and sensitive information, our released source code includes a personally identifiable information (PII) detector that automatically identifies and redacts PII before data storage. We strongly discourage users from targeting sensitive or private content.
>
> **Copyright and Legal Compliance**: The system provides warnings when encountering potentially copyrighted material and maintains detailed logs through our OHCache system to ensure data traceability and accountability.
>
> Although the aforementioned features may hinder AutoData in performance, cost efficiency, or robustness, we prioritize ethical practices over comprehensive coverage. We strictly follow the NeurIPS Code of Ethics and regional legislations, i.e., never attempting to break security measures, only accessing content the user has legitimate rights to access, respecting rate limits, and server resources. Notice that the objective of Autodata is to automate **legitimate open web data collection**, not to circumvent security measures.

---

> > ### Comment · Reviewer_ddG4 · 2025-08-02
> >
> > Thank you for your response. Your clarifications addressed my concerns, and I have no further questions at this point.

---

> > > ### Author Response · Authors · 2025-08-03
> > >
> > > Thank you once again for reviewing our work. Your feedback is valuable to our study!

---

### Decision · Program_Chairs · 2025-09-17

**Decision:**

Accept (poster)

**Comment:**

**summary**
This paper introduces AutoData, a multi-agent system designed to automate web data collection using only a natural language instruction. The system's core innovation is the Oriented Hypergraph Cache (OHCache), a communication architecture that enables efficient, structured messaging and reduces the high token costs associated with LLM-based agents. The authors also contribute Instruct2DS, a new benchmark for evaluating live data collection tasks across multiple domains.

**strengths**
- The paper addresses the highly practical and relevant problem of automated web data collection, offering a novel system that significantly reduces manual effort.
- The proposed OHCache architecture is an effective design for optimizing communication in multi-agent systems, successfully reducing token usage and execution time compared to baselines.
- The introduction of the Instruct2DS benchmark is a valuable contribution, providing a much-needed resource for standardized and realistic evaluation of web scraping agents.

**weaknesses**
- The system's robustness on complex websites that rely heavily on JavaScript, require logins, or use CAPTCHAs is not fully demonstrated, limiting its real-world applicability.
- The current implementation is limited to text-centric data and does not support multimodal data collection (e.g., images, video), which is a significant part of web content.
- The system's best performance is dependent on expensive proprietary models like GPT-4o, and the path to achieving similar results with open-weight models remains future work.

**final descision**
Accept. The paper presents a well-designed system for an important problem, and its contributions, particularly the novel OHCache architecture and the Instruct2DS benchmark, are significant and well-supported by strong empirical results.